# WHIM Syndrome-linked *CXCR4* mutations drive osteoporosis

Adrienne Anginot[1,2,3,11], Julie Nguyen[2,4,11], Zeina Abou Nader[1,2,3,11], Vincent Rondeau[1,2,3], Amélie Bonaud[1,2,3], Maria Kalogeraki[1,2,3], Antoine Boutin[5], Julia P. Lemos[1,2,3], Valeria Bisio[1,2,3], Joyce Koenen[2,4], Lea Hanna Doumit Sakr[6], Amandine Picart [6], Amélie Coudert[6], Sylvain Provot [6], Nicolas Dulphy [1,2,3], Michel Aurrand-Lions [2,3,7], Stéphane J. C. Mancini [2,7], Gwendal Lazennec [2,8], David H. McDermott [9], Fabien Guidez[3,10], Claudine Blin-Wakkach [5], Philip M. Murphy[9], Martine Cohen-Solal [6], Marion Espéli [1,2,3], Matthieu Rouleau [5] & Karl Balabanian [1,2,3] ✉

WHIM Syndrome is a rare immunodeficiency caused by gain-of-function *CXCR4* mutations. Here we report a decrease in bone mineral density in 25% of WHIM patients and bone defects leading to osteoporosis in a WHIM mouse model. Imbalanced bone tissue is observed in mutant mice combining reduced osteoprogenitor cells and increased osteoclast numbers. Mechanistically, impaired CXCR4 desensitization disrupts cell cycle progression and osteogenic commitment of skeletal stromal/stem cells, while increasing their pro-osteoclastogenic capacities. Impaired osteogenic differentiation is evidenced in primary bone marrow stromal cells from WHIM patients. In mice, chronic treatment with the CXCR4 antagonist AMD3100 normalizes in vitro osteogenic fate of mutant skeletal stromal/stem cells and reverses in vivo the loss of skeletal cells, demonstrating that proper CXCR4 desensitization is required for the osteogenic specification of skeletal stromal/stem cells. Our study provides mechanistic insights into how CXCR4 signaling regulates the osteogenic fate of skeletal cells and the balance between bone formation and resorption.

The bone marrow (BM) is a complex structural and primary immune organ whose development and maintenance depend on multiple cell types including cells of the hematopoietic lineage like hematopoietic stem and progenitor cells (HSPCs), but also vascular cells and numerous skeletal cells encompassing BM stromal cells (BMSCs), skeletal progenitor/precursor cells as well as bone-making osteoblasts (OBLs)[1,2]. Together these cells compose specialized micro-anatomical structures called "niches" that sustain their survival and differentiation[3–9]. For instance, the HSPC niches are thought to be composed of perivascular stromal units associated with sinusoids and arterioles[10–15]. Bone and adipose cells are thought to derive from subsets of BMSCs that are located near blood vessels and function as

[1]Université Paris Cité, Institut de Recherche Saint-Louis, INSERM U1160 Paris, France. [2]CNRS, GDR3697 "Microenvironment of tumor niches", Micronit, France. [3]OPALE Carnot Institute, The Organization for Partnerships in Leukemia, Hôpital Saint-Louis, Paris, France. [4]Inflammation, Microbiome and Immunosurveillance, INSERM, Université Paris-Saclay, Orsay, France. [5]Université Côte d'Azur, CNRS, LP2M Nice, France. [6]Université Paris Cité, BIOSCAR Inserm U1132, Department of Rheumatology and Reference Center for Rare Bone Diseases, AP-HP Hospital Lariboisière, Paris, France. [7]Aix Marseille Univ, CNRS, INSERM, Institut Paoli-Calmettes, CRCM, Marseille, France. [8]CNRS, SYS2DIAG-ALCEDIAG, Cap Delta, Montpellier, France. [9]Molecular Signaling Section, Laboratory of Molecular Immunology, National Institute of Allergy and Infectious Diseases, NIH, Bethesda, MD, USA. [10]Université Paris Cité, Institut de Recherche Saint-Louis, INSERM U1131 Paris, France. [11]These authors contributed equally: Adrienne Anginot, Julie Nguyen, Zeina Abou Nader. ✉e-mail: karl.balabanian@inserm.fr

skeletal stromal/stem cells (SSCs)[16–19]. However, the exact localization, composition and crossover of these niches in relation with bone function are not yet established. Bone tissue homeostasis relies on the balance between formation and resorption of bone matrix mediated by effector cells that derive from SSCs and HSPCs, respectively. Disequilibrium of this balance can lead to diseases such as osteoporosis or osteopetrosis. In such a landscape, SSCs are key players: not only they give rise to OBLs but they also contribute to perivascular structures important for HSPCs[20–28]. Understanding how SSCs maintain their identity, achieve plasticity and support hematopoiesis in adult BM is thus an important emerging field[3,6,9,12,29]. Recently, Ambrosi and coll. showed that intrinsic ageing of SSCs skews skeletal and hematopoietic lineage outputs, leading to fragile bones[30]. Moreover, Jeffery and coll. reported how BM and periosteal SSCs contribute to bone maintenance and repair[31]. However, both extrinsic and intrinsic mechanisms regulating their fate remain incompletely understood.

In adult BM, signaling by the G protein-coupled receptor CXCR4 on HSPCs in response to stimulation by the chemokine CXCL12/ Stromal cell-derived factor-1, produced by BMSCs, constitutes a key pathway through which the stromal niches and HSPCs communicate[32–37]. Conditional ablation of *Cxcl12* from perivascular stromal cells or OBLs demonstrated that HSCs occupy a perivascular but not an endosteal niche[21,38], whereas targeted deletion of *Cxcl12* from BM stromal cells has allowed the identification of specialized niches supporting leukemia stem cell maintenance[39]. Both Cxcr4 and Cxcl12 are broadly expressed by non-hematopoietic tissues and cell types and have multifunctional roles beyond hematopoiesis. Since mice deficient for *Cxcr4* or *Cxcl12* die perinatally, our understanding of the role of the Cxcl12/Cxcr4 axis in regulating the BM ecosystem is mostly based on relatively selective loss-of-function models[21,40–44]. Conditional inactivation of *Cxcl12* or *Cxcr4* in paired-related homeobox gene 1 (Prx1)- or osterix (Osx)-expressing cells, i.e., respectively, multipotent mesenchymal progenitors or osteoprogenitor cells (OPCs) and descendant OBLs, was associated with reduced postnatal bone formation, suggesting a positive regulatory role of this pair in OBL development and/or function[21,43,44]. Single cell transcriptomics recently suggested heterogeneity within adult Cxcl12- or Leptin-receptor (LepR)-expressing mesenchymal cells poised to undergo either adipogenic or osteogenic specification[45,46]. However, it is still unclear whether and how Cxcr4 signaling regulates intrinsically osteogenic specification of skeletal cells.

Here, we addressed this point using as a paradigm the WHIM Syndrome (WS), a rare immunodeficiency caused by viable inherited heterozygous gain-of-function mutations in *CXCR4* affecting homologous desensitization of the receptor, thus resulting in enhanced signaling following CXCL12 stimulation, defective lymphoid differentiation of HSPCs and reduced blood leukocyte numbers[47–50]. Taking advantage of a mouse strain that harbors the naturally occurring WS-linked heterozygous *CXCR4^S338X* mutation (*Cxcr4^{+/1013}*, +/1013)[51–54], and of human BM samples from WS donors and clinical data from 19 WS patients, we investigated whether WS mutations affect the SSC landscape with a particular attention given to the bone fraction. WS-linked *CXCR4* mutations were associated with reduced bone mass in mice and humans. In mice, this relied on impaired CXCR4 desensitization that disrupts cell cycle progression and osteogenic commitment of SSCs, while paradoxically increasing the pro-osteoclastogenic capacities of these cells. Impairment in osteogenic potential was also evidenced in BMSCs from WS patients. Thus, proper CXCR4 desensitization is required for the osteogenic specification of bone SSCs.

## Results
### WS-linked *CXCR4* mutations are associated with reduced bone mass in mice and humans
Following CXCL12 stimulation, β-arrestins are recruited to the carboxyl-terminal tail (C-tail) domain of CXCR4, precluding further

## Table 1 | Abnormal bone mineral density values in WS patients

| | Gender | Chronic treatment | Lumbar spine | Femoral neck |
|---|---|---|---|---|
| P1 | Female | No | −3.1[a] | 0 |
| P2 | Female | No | −1.1 | −1.8 |
| P3[b] | Male | Yes[c] | −1.8 | −2.3 |
| P4 | Female | Yes[d] | −2.7 | −1.3 |
| P5[b] | Male | Yes[e] | −1.8 | −2.2 |

Characteristics of each patient with low BMD value are shown. Four patients carry the *CXCR4^R334X* mutation and one displays the *CXCR4^S338X* mutation. There were 3 women and 2 men with an average age of 33.2 years (range 13–52). T-scores for lumbar spine (L1-L4) and femoral neck have been evaluated. According to World Health Organization (WHO) criteria, values classify patients as osteopenic with a T-score between −1.0 and −2.5 or osteoporotic with a T-score at or below −2.5.
[a]Values outside the normal range defined by WHO are italicized.
[b]For patients 3 and 5, because of their young age, Z-scores are given with a value at or below −2.0 considered as abnormal.
[c]G-CSF since age of 2.
[d]G-CSF several years at the time of scan.
[e]G-CSF for 6 months at the time of scan.

G-protein activation (i.e., desensitization) and leading to receptor internalization[55]. Both processes are dysregulated in WS most often due to autosomal-dominant gain-of-function mutations that result in the distal truncation of the C-tail of CXCR4 and a desensitization-resistant, hyperactive receptor[56]. Although the impact of these WS mutations on immune cells is currently being understood[51–54,57], nothing is known about their impact on the skeletal landscape. Bone mineral density (BMD) values were measured in 19 patients with WS for lumbar spine and femoral neck by total body dual-energy X-ray absorptiometry. BMD T- and Z-scores were found to be low at least in one site in five patients (Table 1). Likewise, this was evidenced in adult (8–12-week-old) *Cxcr4^{1013}*-bearing (i.e., heterozygous [+/1013] and homozygous [1013/1013]) mice, as compared to *Cxcr4^{+/+}* (WT) mice. Analyses of lumbar spine revealed decreased BMD values in mutant mice, in a *Cxcr4^{1013}* allele dose-dependent manner (Fig. 1A). Micro-computed tomography (μCT) analyses further unraveled a reduction in trabecular and cortical bone in mutant mice (Fig. 1B). In mutant femurs, there was a reduction in the trabecular bone density that followed a *Cxcr4^{1013}* allele copy number-dependent pattern. This was characterized by a significant decrease in bone volume and trabecular numbers, while the trabecular separation was increased compared to WT mice (Fig. 1C). The cortical bone volume and thickness were also affected (Fig. 1B, D). This gene-dependent reduction was observed among both female and male mutant mice. Histomorphometric analyses confirmed decreased bone volume and trabecular numbers in mutant mice as shown by toluidine blue and osteopontin (Opn) staining (Fig. 1E, F). Strikingly, staining for Alcian Blue and perilipin that are used for chondrocyte and adipocyte identification, respectively, were unaltered in mutant bone (Fig. 1G, H). Consistently, the thickness of the growth plate as well as the adipocyte content were similar in mice carrying the *Cxcr4* mutation compared to WT ones (Fig. 1I, J), thus suggesting that Cxcr4 specifically regulates the osteogenic fate in adults. Moreover, adult mutant mice did not exhibit significant changes of body size or weight (Fig. 1K), indicating that the *Cxcr4* mutation likely does not alter the skeletal growth. Overall, these findings revealed an osteopenic skeleton in *Cxcr4^{1013}*-bearing mice and low BMD in 25% of WS patients.

### Reduction of skeletal stromal cells in *Cxcr4^{1013}*-bearing mice
We then evaluated by flow cytometry the bone composition of WT and mutant mice with a focus on skeletal cells that encompass notably SSCs, OPCs, and OBLs[58]. Long bones were flushed and then digested. Total stromal cells in the bone fraction were identified as negative for CD45, Lineage (including Ter119), c-Kit, and CD71 expression as

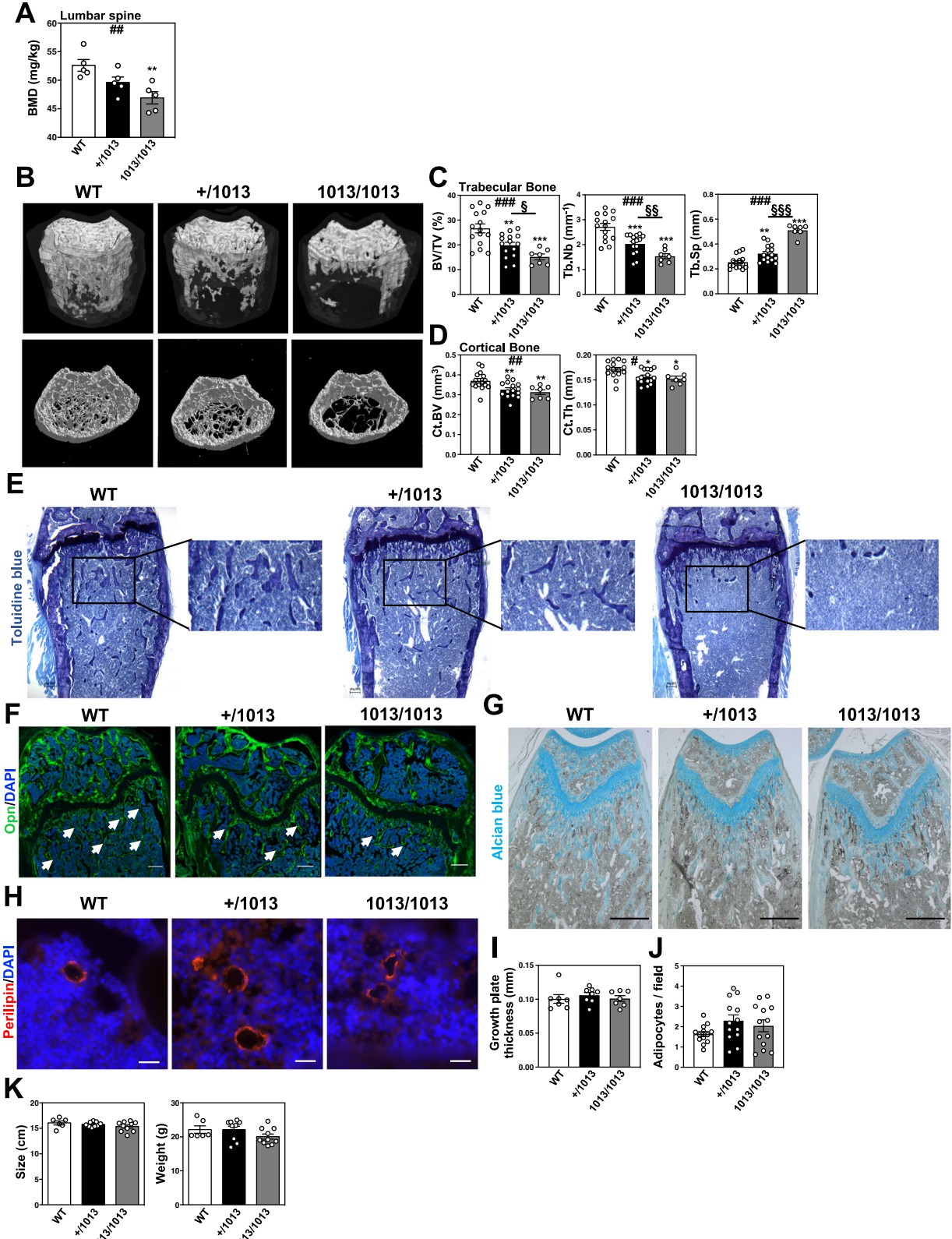

previously reported[12,59]. Endothelial cells were excluded based on CD31 expression. Two distinct CD51+ stromal cell subsets were identified based on Sca-1 and PDGFRα: SSCs (Sca-1+PDGFRα+) and more differentiated OPCs (Sca-1-PDGFRα+/-) (Fig. 2A). We consider Sca-1+PDGFRα+ cells at the top of the hierarchy based on previous works that indicated a highest CFU-F clonogenic potential of these cells as compared to Sca-1-PDGFRα+ cells[12,28,31,60]. We observed a significant decrease of the numbers of OPCs, and to a lesser extent of SSCs, that followed a

Cxcr4[1013] allele dose-dependent pattern (Fig. 2B), thus reinforcing that the landscape of the bone stroma is altered in Cxcr4[1013]-bearing mice.

We next examined in vitro the function of the signaling trio formed by Cxcl12 and its two receptors Cxcr4 and Ackr3 in skeletal cells. Membrane expression of Cxcr4 and Ackr3 was similar between WT and mutant skeletal cells including SSCs (Fig. S1A, S1B). However, +/1013 and 1013/1013 SSCs displayed both impaired Cxcr4 internalization following Cxcl12 stimulation as well as increased Cxcl12-

**Fig. 1 | WS-linked *Cxcr4* mutations are associated with reduced bone mass in mice. A** The bone mineral density (BMD) of lumbar spine of WT, +/1013, and 1013/1013 mice was measured through Dual-energy x-ray absorptiometry. Results represent means ± SEM with 5 mice per group examined over two independent experiments. Statistics were calculated with the nonparametric Kruskal–Wallis H test ($^{\#\#}p = 0.0042$) and the nonparametric Mann–Whitney test, two-sided, +/1013 $p = 0.0556$, 1013/1013 $^{**}p = 0.0079$. **B**–**D** 3D representative images of trabecular and cortical composites (**B**) and quantitative μCT analyses of trabecular (**C**) and cortical (**D**) parameters of femurs from WT and mutant mice. BV = bone volume; TV = trabecular volume; Tb.Nb = trabecular number; Tb.Sp = trabecular separation; Ct.BV = cortical bone volume; Ct.Th = cortical thickness. Data (means ± SEM) are from three independent experiments with n = 15, 15, and 7 mice in total for WT, +/1013, and 1013/1013 groups, respectively. Statistics were calculated with the nonparametric Kruskal–Wallis H test ($^{\#\#\#}p = 0.0007$, BV/TV; $p = 0.0001$, Tb.Nb; $p < 0.0001$, Tb.Sp; Ct.BV, $^{\#\#}p = 0.0072$; $^{\#}p = 0.0241$) and the unpaired two-tailed Student's t test (+/1013 vs WT $^{**}p = 0.0556$, 1013/1013 vs WT $^{***}p = 0.0005$, +/1013 vs 1013/1013 $^{\$}p = 0.032$ for BV/TV; $^{***}p = 0.0005$, $^{***}p < 0.0001$, $^{\$\$}p = 0.008$ for Tb.Nb; $^{***}p = 0.0038$, $^{***}p < 0.0001$, $^{\$\$\$}p < 0.0001$ for Tb.Sp; $^{**}p = 0.0072$, $^{**}p = 0.0078$ for Ct.BV; $^{*}p = 0.0147$, $^{*}p = 0.02$ for Ct.Th). **E** BM sections were stained with toluidine blue. Larger images show 2X inserts in trabecular areas. Bars: 200 μm. Images are representative of at least three independent determinations. **F** BM sections were immunostained for osteopontin (Opn) in association with DAPI. Trabeculae are indicated by white arrows. Bars: 250 μm. Images are representative of five independent determinations. **G, H** BM sections were stained for chondrocyte (alcian blue, **G**) or adipocyte (perilipin, **H**) markers. Bars: 500 (**G**) or 20 (**H**) μm. Images are representative of at least six independent determinations. **I** Cartilaginous growth plates were evaluated based on overall growth plate thickness measured on μCT scans. Data (means ± SEM) are from 2 independent experiments with n = 7, 8, and 7 mice in total for WT, +/1013, and 1013/1013 groups, respectively. **J** Adipocyte counts were evaluated on perilipin-stained BM sections. Six fields of 3 mm² were analyzed per section. Results (means ± SEM) are from 4 independent experiments with 13 mice in total per group. **K** Size (left) and weight (right) of WT and mutant mice. Results (means ± SEM) are from five independent experiments with n = 6, 9, and 10 mice in total for WT, +/1013, and 1013/1013 groups, respectively. Mice were littermates, females, and age-matched (8–12 wk-old) in (**A**–**J**) and at 8 weeks of age in (**K**). Source data are provided as a Source data file.

mediated chemotaxis that was abolished by the specific Cxcr4 antagonist AMD3100 (Fig. S1C, S1D). These dysfunctions likely relied on the enhanced signaling properties of the truncated Cxcr4 receptor as revealed by Erk PhosphoFlow analyses (Fig. S1E). Combined with the apparent preserved capacity of Ackr3 to bind and internalize Cxcl12 in vitro (Fig. S1F, S1G), these findings indicated a functional expression of the desensitization-resistant C-tail-truncated Cxcr4$^{1013}$ receptor on SSCs. Abnormal Cxcr4 signaling was not associated with changes in apoptosis of SSCs (Fig. S1H).

To determine whether reduction of bone content in *Cxcr4$^{1013}$*-bearing mice resulted from defects intrinsic to skeletal cells and/or an alteration of the hematopoietic (or another non-stromal) system, we performed reciprocal long (16 weeks)- and short (3 weeks)-term BM reconstitution experiments. First, BM cells from WT CD45.1$^{+}$ mice were transplanted into lethally irradiated 8-wk-old CD45.2$^{+}$ WT or mutant (+/1013 and 1013/1013) mice (Fig. 2C). Sixteen weeks later, mutant recipients exhibited CD45.1$^{+}$ chimerism in hematopoietic compartments similar to those of WT recipients (Fig. 2D) but displayed reduced numbers of skeletal cells including SSCs and OPCs (Fig. 2E). Confocal imaging and μCT analyses confirmed that transplantation of WT BM was not sufficient to rescue the trabecular network in mutant recipients (Fig. 2F–I). This was also evidenced three weeks after WT BM transplantation (Fig. 2G). Although we cannot formally exclude that WT hematopoietic cells may be able to rescue the reduced trabecular bone content in mutant mice earlier in development, these results suggested that skeletal cell-autonomous Cxcr4 regulation contributes to the persistent bone defects in adult *Cxcr4$^{1013}$*-bearing mice. We then performed reverse chimeras in which irradiated 8-wk-old CD45.1$^{+}$ WT mice were reconstituted with WT, +/1013 or 1013/1013 CD45.2$^{+}$ BM (Fig. 3A). Sixteen weeks later, CD45.2$^{+}$ chimerism of LT-HSCs and leukocytes were decreased, respectively, in BM and blood of CD45.1$^{+}$ WT recipients engrafted with mutant BM, confirming the impaired reconstitution capacity of mutant HSCs (Fig. 3B and ref. 49). There were significantly lower numbers of skeletal cells and defective trabecular bone content in *Cxcr4$^{1013}$*-bearing BM-chimeric mice compared to WT chimeras as early as 3 weeks post-transplantation (Fig. 3C–G), thereby indicating cell-extrinsic Cxcr4-mediated regulation of the skeletal landscape. Likewise, cortical bone content was slightly affected (Fig. 3H, I). Altogether, these findings suggest that impaired Cxcr4 desensitization in both skeletal and hematopoietic cells have combinatorial effects on bone landscape dysregulation in adult *Cxcr4$^{1013}$*-bearing mice.

## Increased bone resorption and reduced bone formation in *Cxcr4$^{1013}$*-bearing mice

Bone is maintained by coupled activities of bone-forming OBLs and bone-resorbing osteoclasts (OCLs). Alterations in bone balance can result in pathologic bone loss and osteoporosis. This led us to investigate whether the gain-of-*Cxcr4*-function mutation modulates the OBL/OCL balance. First, we analyzed bone resorption by quantifying OCL numbers in mice using Tartrate Resistant Acid Phosphatase (TRAP) staining[61]. We observed increased OCL surface (Oc.S/BS) and number (Oc.N/BV) in mutant mice compared to WT ones (Fig. 4A, B). To determine whether the increased bone resorption in mutant mice resulted from OCL-intrinsic defects, we performed in vitro OCL differentiation from BM cells in the presence of M-Csf and Rank-L and tested their bone resorption capacity. Similar OCL numbers and bone matrix resorption activities were observed among WT and mutant cultures (Fig. S2A, S2B), suggesting preserved intrinsic capacities of mutant BM myeloid cells to differentiate in vitro into functional OCLs. Congruent with this, we observed no changes in expression levels of osteoclastogenic genes in mutant cultures compared to WT ones (Fig. S2C). These findings indicate that the *Cxcr4* mutation does not affect in vitro OCL differentiation and function but suggest that osteoclastogenesis and increased bone resorption in mutant mice may be promoted by the BM environment.

*Cxcr4$^{1013}$*-bearing mice exhibited similar bone formation as revealed by osteoid surface (OS/BS) and osteoblast surface (Obl.S/BS) compared to WT mice (Fig. 4C). Dynamic parameters of in vivo bone formation were also assessed by quantifying bone surfaces labeled with tetracycline and calcein. Total and double-labeled surfaces were lower in mutant than WT mice (Fig. 4D), while mineral apposition rate (MAR) were similar in WT and *Cxcr4$^{1013}$*-bearing mice (Fig. 4E). Overall, bone formation rate (BFR/BS) was significantly reduced in mutant mice (Fig. 4E). This suggests a decrease in bone formation related to a lower number of OPCs and OBLs with maintained activity of individual OBL. In line with preserved intrinsic bone formation capacities of active osteoblastic lineage cells in mutant mice, high-throughput RNA sequencing (RNA-seq) and qPCR analyses of OPC bulk-sorted by flow cytometry from the bone fraction on the basis of the CD51 and Sca-1 markers (Fig. 2A) highlighted a gene signature with preserved mineralized matrix potential in mutant OPCs (Figs. 4F–H and S2D–F). In agreement, sorted OPCs from mutant mice were as efficient as WT ones in vitro at producing differentiated OBLs and mineralized nodules after 14- or 21-days culture in osteogenic medium as determined by Alkaline phosphatase (Alp) and Alizarin Red (AR) staining, respectively[24,60,62,63] (Figs. 4I and S2G). This was confirmed by qPCR analyses with no changes in expression of genes encoding osteogenic regulators in mutant cultures (Fig. S2H). These findings suggest reduced osteogenic lineage commitment in *Cxcr4$^{1013}$*-bearing mice. Given that osteogenic cells support osteoclastogenesis through the production of factors such as Rank-L (*Tnfsf11*)[64], we questioned our RNAseq data on the related gene expression profile in mutant

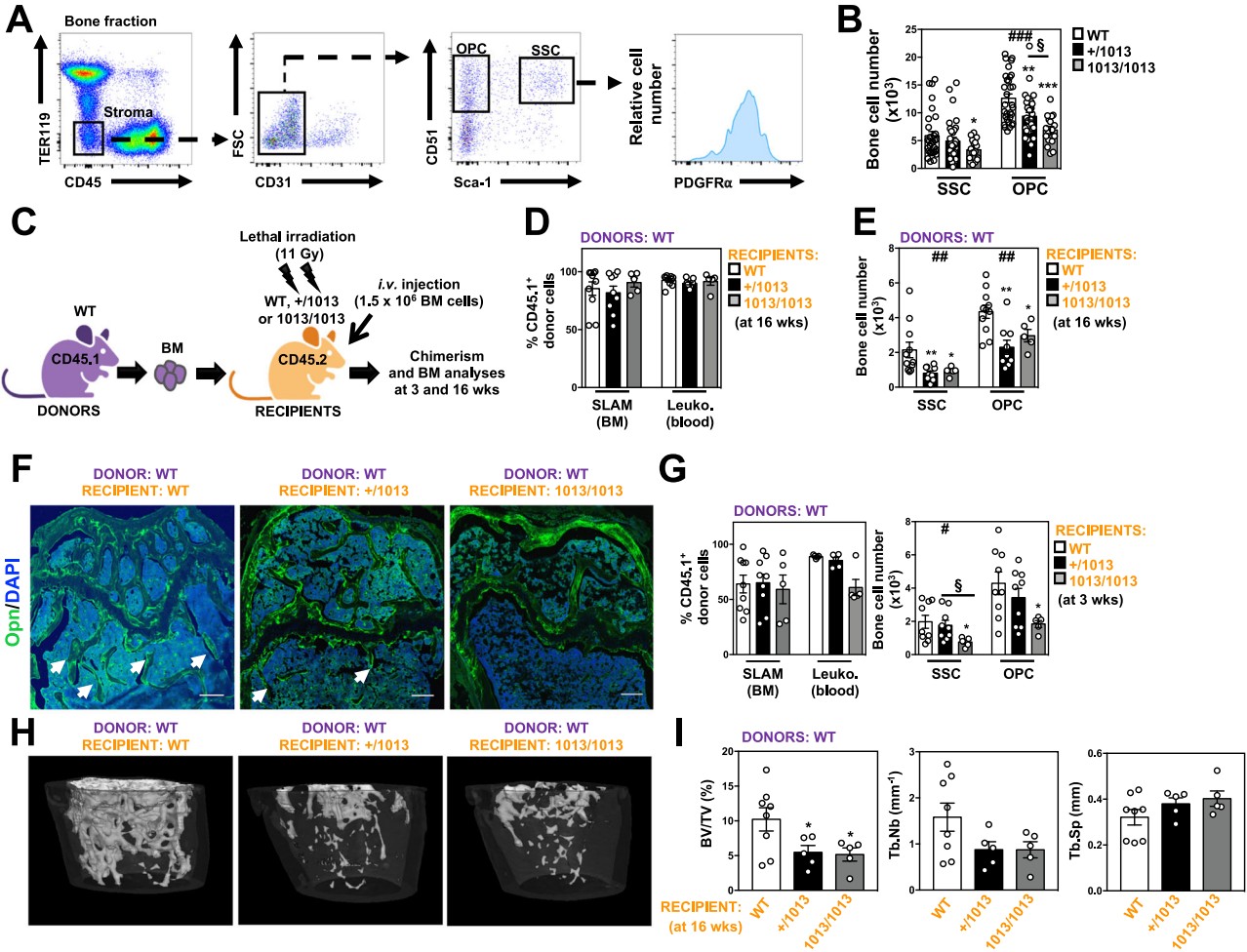

**Fig. 2 | Reduction of skeletal stromal cells in *Cxcr4^{1013}*-bearing mice.**
**A** Representative dot-plots showing the flow cytometric gating strategies used to sort stroma cells (CD45⁻TER119⁻), differentiated osteoblast progenitor cells (OPCs, CD45⁻TER119⁻CD31⁻Sca-1⁻CD51⁺PDGFRα⁺/⁻) and SSCs (CD45⁻TER119⁻CD31⁻Sca-1⁺CD51⁺PDGFRα⁺) in the mouse bone fraction. **B** Absolute numbers of the indicated stroma cell subsets from bone fractions were determined by flow cytometry. Data (means ± SEM) are from at least six independent experiments with $n$ = 31, 31, and 17 mice in total for WT, +/1013, and 1013/1013 groups, respectively. Statistics were calculated with the nonparametric Kruskal–Wallis H test (###$p$ < 0.0001, OPC) and the unpaired two-tailed Student's t test (1013/1013 vs WT *$p$ = 0.0206 for SSC; +/1013 vs WT **$p$ = 0.0031, 1013/1013 vs WT ***$p$ < 0.0001, +/1013 vs 1013/1013 §$p$ = 0.0217 for OPC). **C** Schematic diagram for the generation of CD45.1→CD45.2 short (3 wks)- or long (16 wks)-term BM chimeras. Mouse icons were created using the Biorender software (Biorender.com, agreement number: VX255VH9TZ). **D** Proportions of WT donor CD45.1⁺ LSK SLAM and leukocytes (Leuko.) recovered from the BM and blood of BM chimeras in CD45.2⁺ WT or mutant recipients 16 weeks after transplantation. **E** Absolute numbers of SSCs and OPCs determined by flow cytometry in bone fractions of BM chimeras in CD45.2⁺ recipients. Data (means ± SEM) in (**D**) and (**E**) are from three independent experiments with $n$ = 10 (D) or 11 (E), 9, and 5 mice in total for WT, +/1013, and 1013/1013 recipient groups, respectively. Statistics were calculated with the nonparametric Kruskal–Wallis H test (##$p$ = 0.0064) and the nonparametric Mann–Whitney test, two-sided (+/1013 vs WT **$p$ = 0.0042, 1013/1013 vs WT *$p$ = 0.0126, for SSC; +/1013

vs WT **$p$ = 0.0031, 1013/1013 vs WT *$p$ = 0.0398, for OPC). **F** Sixteen weeks after transplantation, BM sections from WT or mutant CD45.2⁺ recipient mice were immunostained for Opn in association with DAPI (bars: 250 μm). Trabeculae are indicated by white arrows. Images are representative of at least three independent determinations. **G** Left: Proportions of WT donor CD45.1⁺ LSK SLAM and leukocytes recovered from the BM and blood of BM chimeras in CD45.2⁺ WT or mutant recipients 3 weeks after transplantation. Right: Absolute numbers of SSCs and OPCs. Data (means ± SEM) are from three independent experiments with $n$ = 9, 9, and 5 mice in total for WT, +/1013, and 1013/1013 recipient groups, respectively, except for blood chimerism analysis (5 mice per group). Statistics were calculated with the nonparametric Kruskal–Wallis H test (#$p$ = 0.0327) and the nonparametric Mann–Whitney test, two-sided (1013/1013 vs WT *$p$ = 0.019, +/1013 vs 1013/1013 §$p$ = 0.019 for SSC; 1013/1013 vs WT *$p$ = 0.017 for OPC). **H, I** 3D representative images of trabecular composites (**H**) and μCT analyses of trabecular parameters (**I**) of femurs from WT BM-chimeric CD45.2⁺ WT or mutant recipients 4 months after transplantation. Data (means ± SEM) are from three independent experiments with $n$ = 8, 5, and 5 mice in total for WT, +/1013, and 1013/1013 recipient groups, respectively. Statistics were calculated with the unpaired two-tailed Student's t test (+/1013 vs WT *$p$ = 0.033, 1013/1013 vs WT *$p$ = 0.024 for BV/TV). WT and mutant mice were littermates, females and age-matched (8–12 wk-old) and adult Boy/J (CD45.1) WT mice at 8 weeks of age were used as BM donors. Source data are provided as a Source data file.

and WT OPCs. No major changes in expression levels of pro-osteoclastogenic or anti-resorptive genes were revealed in mutant OPCs (Fig. 4J, K).

To investigate whether increase in OCLs in mutant mice is linked to an altered stromal BM environment, we sought to set up co-culture experiments between in vitro-expanded osteogenic cells carrying or not the *Cxcr4* mutation and WT OCL precursors, i.e., BM CD11b⁺

myeloid cells as reported[65]. We showed that mutant osteogenic cells promoted exacerbated OCL differentiation compared to WT cells (Fig. 4L). Soluble factors seem not to be sufficient as the supernatants of such stimulated expanded osteogenic cells (WT or mutant) did not induce OCL differentiation. Additionally, transcriptomic analyses of stimulated osteogenic cells carrying or not the *Cxcr4* mutation did not reveal any major changes in expression levels of master genes

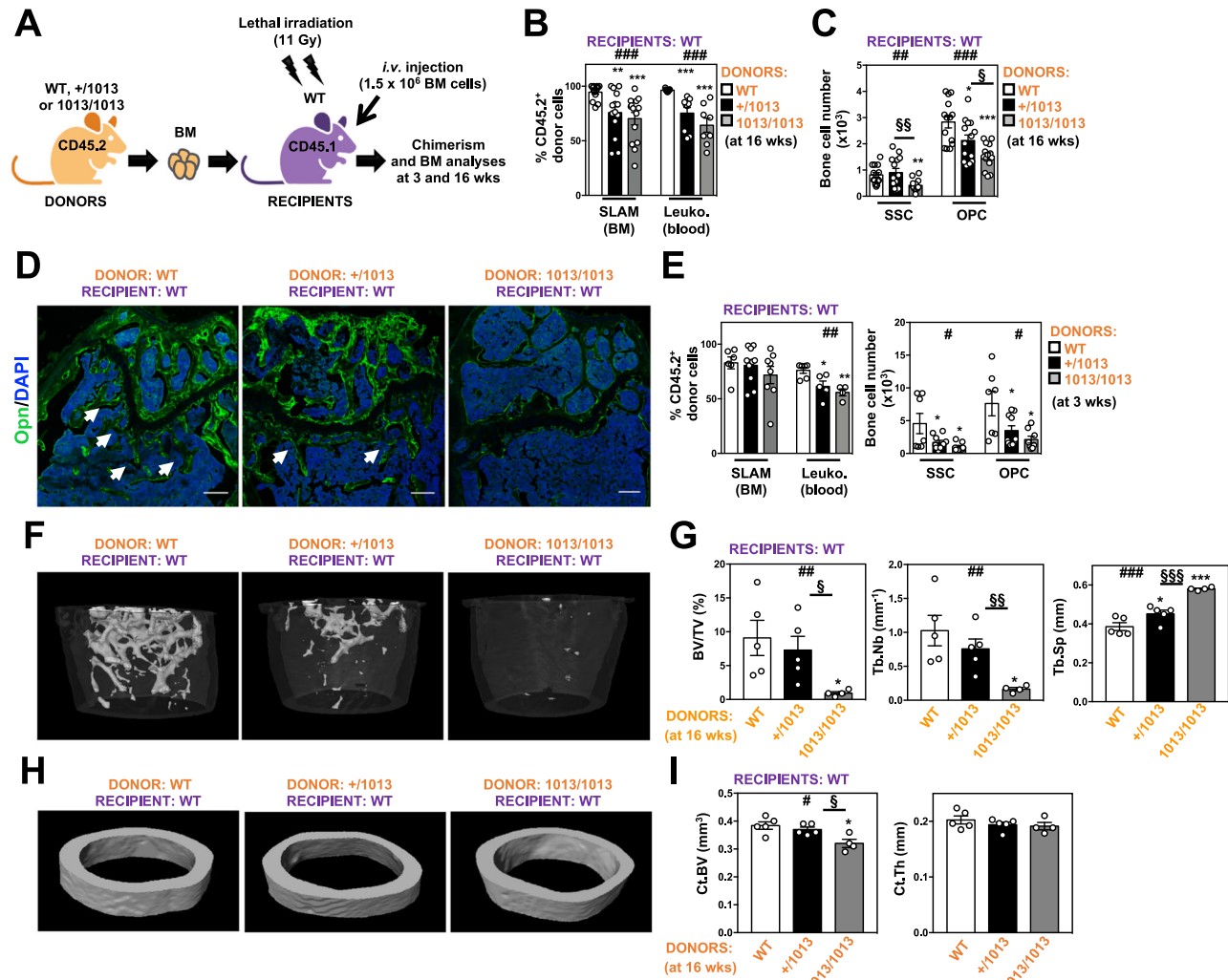

**Fig. 3 | Cell-extrinsic Cxcr4-mediated regulation of the skeletal landscape.**
**A** Schematic diagram for the generation of CD45.2→CD45.1 short (3 wks)- or long (16 wks)-term BM chimeras. Mouse icons were created using the Biorender software (Biorender.com, agreement number: VX255VH9TZ). **B** Proportions of WT or mutant donor CD45.2+ LSK SLAM and leukocytes recovered from the BM and blood of BM chimeras in CD45.1+ WT recipients 16 weeks after transplantation. Statistics were calculated with the nonparametric Kruskal–Wallis H test (###p = 0.0008 for SLAM and <0.0001 for Leukocytes) and the unpaired two-tailed Student's t test (+/1013 vs WT **p = 0.0036, 1013/1013 vs WT ***p = 0.0004, for SLAM; +/1013 vs WT ***p = 0.0003, 1013/1013 vs WT ***p < 0.0001, for Leukocytes). **C** Absolute numbers of SSCs and OPCs in bone fractions of BM chimeras in CD45.1+ recipients. Statistics were calculated with the nonparametric Kruskal–Wallis H test (##p = 0.009 for SSC and ###p = 0.0006) and the unpaired two-tailed Student's t test (1013/1013 vs WT **p = 0.0012, +/1013 vs 1013/1013 §§p = 0.0076, for SSC; +/1013 vs WT *p = 0.029, 1013/1013 vs WT ***p < 0.0001, +/1013 vs 1013/1013 §p = 0.022, for OPC). Data (means ± SEM) in (**B**) and (**C**) are from three independent experiments with n = 14 mice in total for WT, +/1013, and 1013/1013 donor groups, respectively, except for blood chimerism analysis (n = 11, 9, and 8 mice in total for WT, +/1013, and 1013/1013 donor groups, respectively). **D** Sixteen weeks after transplantation, BM sections from WT CD45.1+ recipient mice were immunostained for Opn in association with DAPI (bars: 250 μm). Trabeculae are indicated by white arrows. Images are representative of at least three independent determinations. **E** Left: Proportions of WT or mutant donor CD45.2+ LSK SLAM and leukocytes recovered from the BM and blood of BM chimeras in CD45.1+ WT recipients 3 weeks after transplantation. Right:

Absolute numbers of SSCs and OPCs. Data (means ± SEM) are from three independent experiments with n = 6 (SLAM) or 7 (SSC and OPC), 10, and 8 mice in total for WT, +/1013, and 1013/1013 donor groups, respectively, except for blood chimerism analysis (n = 6, 5, and 4 mice in total for WT, +/1013, and 1013/1013 donor groups, respectively). Statistics were calculated with the nonparametric Kruskal–Wallis H test (##p = 0.0083 for leukocytes; #p = 0.047 for SSC; #p = 0.018 for OPC) and the unpaired two-tailed Student's t test (+/1013 vs WT *p = 0.021, 1013/1013 vs WT **p = 0.0014, for leukocytes; +/1013 vs WT *p = 0.046, 1013/1013 vs WT *p = 0.029, for SSC; +/1013 vs WT *p = 0.034, 1013/1013 vs WT *p = 0.01, for OPC). **F–I** 3D representative images of trabecular or cortical composites (**F** and **H**) and μCT analyses of trabecular or cortical parameters (**G** and **I**) of femurs from WT or mutant BM-chimeric CD45.1+ WT recipients 4 months after transplantation. Ct.BV = cortical bone volume; Ct.Th = cortical thickness. Data (means ± SEM) in (**G**) and (**I**) are from two independent experiments with n = 5, 5, and 4 mice in total for WT, +/1013, and 1013/1013 donor groups, respectively. Statistics were calculated with the nonparametric Kruskal–Wallis H test (##p = 0.0085 for BV/TV; ##p = 0.0069 for Tb.Nb; ###p = 0.0001 for Tb.Sp; #p = 0.033 for Ct.BV) and the unpaired two-tailed Student's t test (1013/1013 vs WT *p = 0.028, +/1013 vs 1013/1013 §p = 0.028 for BV/TV; 1013/1013 vs WT *p = 0.011, +/1013 vs 1013/1013 §§p = 0.0088 for Tb.Nb; +/1013 vs WT *p = 0.044, 1013/1013 vs WT ***p < 0.0001, +/1013 vs 1013/1013 §§§p = 0.0006 for Tb.Sp; 1013/1013 vs WT *p = 0.014, +/1013 vs 1013/1013 §p = 0.016 for Ct.BV). Donor WT and mutant mice and Boy/J (CD45.1) WT recipient mice were females at 8 weeks of age. Source data are provided as a Source data file.

regulating osteoclastogenesis (Fig. 4M). These findings suggest a juxtacrine function of osteogenic cells toward OCL differentiation that likely relies on direct interactions between both cell types and involves the Cxcl12/Cxcr4 axis.

Taken as a whole, our findings suggest that the hematopoietic contribution to bone loss in *Cxcr4^{1013}*-bearing mice likely involves dysregulation of the OCL compartment regardless of their activity. The overall decrease in bone mass in mutant mice may involve modulation

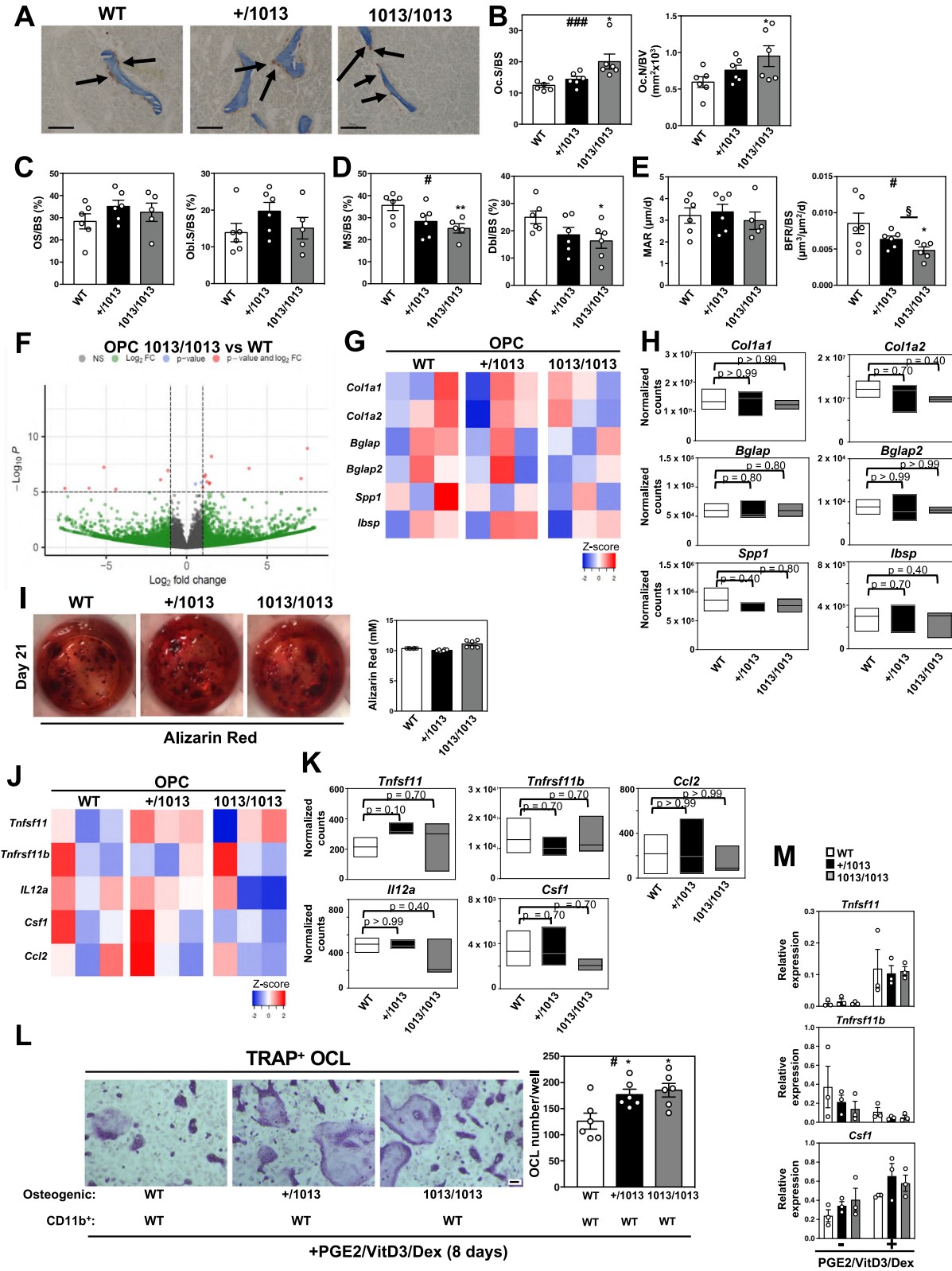

of osteogenic and osteoclastogenic components with decreased bone formation and increased bone resorption. Bone remodeling involves a tight regulation of opposite processes leading to bone resorption and bone formation[66,67]. The observation that immature and mature osteogenic cells, i.e., OPCs and OBLs, displayed preserved intrinsic functions led us to study the early differentiation process of the osteolineage.

## Impaired osteogenic specification of *Cxcr4[1013]*-bearing skeletal stromal cells

We thus investigated the intrinsic characteristics of SSCs carrying the *Cxcr4* mutation. Undifferentiated stem cells are characterized by their slow cell cycle progression in unperturbed conditions[9,68]. This led us to interrogate by flow cytometry the cycling status of SSCs from the bone fractions of *Cxcr4[1013]*-bearing mice by performing DAPI/Ki-67 staining.

**Fig. 4 | Increased bone resorption and reduced bone formation in *Cxcr4^1013*-bearing mice. A** Bone sections were stained for Tartrate Resistant Acid Phosphatase (TRAP) activity (bars: 100 μm). OCLs are visualized as brown-stained TRAP-positive cells attached to bone trabeculae and are indicated by arrows (representative images). **B** OCLs were quantified (Oc.S/BS) and (Oc.N/BV). Results represent means ± SEM with 6 mice in total per group over 3 independent experiments. Statistics were calculated with the nonparametric Kruskal–Wallis H test (###$p = 0.0006$) and the unpaired two-tailed Student's t test (1013/1013 vs WT *$p = 0.012$ for Oc.S/BS; 1013/1013 vs WT *$p = 0.049$ for Oc.N/BV). **C, D** Dynamic histomorphometric measures of bone formation. OS/BS = Osteoid number/Bone surface; Obl.S/BS = Osteoblast surface/Bone surface; MS/BS = Mineralized surface/Bone surface; Dbl/BS = Double-labeled surface/Bone surface. Data (means ± SEM) are from 3 independent experiments with $n = 6$, 6, and 5 mice in total for WT, +/1013, and 1013/1013 groups, respectively, in (**C**), and $n = 6$, 6, and 5 (for MS/BS) or 6 (for Dbl/BS) mice in total for WT, +/1013, and 1013/1013 groups, respectively, in (**D**). Statistics were calculated with the nonparametric Kruskal–Wallis H test (#$p = 0.038$) and the unpaired two-tailed Student's t test (1013/1013 vs WT **$p = 0.0097$ for MS/BS; 1013/1013 vs WT *$p = 0.035$ for Dbl/BS). **E** The mineral apposition rate (MAR) and bone formation rate (BFR/BS) were determined. Results (means ± SEM) are from 3 independent experiments with $n = 6$, 6, and 5 (for MAR) or 6 (for BFR/BS) mice in total for WT, +/1013, and 1013/1013 groups, respectively. Statistics were calculated with the nonparametric Kruskal–Wallis H test (#$p = 0.039$) and the unpaired two-tailed Student's t test (1013/1013 vs WT *$p = 0.03$, +/1013 vs 1013/1013 §$p = 0.044$). **F** Volcano plot analysis of differentially expressed genes obtained by RNA-seq between WT and 1013/1013 OPCs ($p < 0.05$; FC ≥ 2) performed on three biological replicates per group with one replicate representing the pool of 3 mice. Data represent analysis of cpm estimates with a log of fold change of more than 1.5-fold and $p < 0.05$ using enhanced Volcano package. **G, J** Heatmap representing the relative expression levels of selected genes (osteogenic, **G** and osteoclastogenic, **J**) expressed by sorted OPCs. **H, K** Normalized counts of osteogenic (**H**) and osteoclastogenic (**K**) genes using the DESeq2 method. Data are represented as floating bars (min to max and line equal median) of the three biological replicates per group. For significance testing, DESeq2 uses a Wald test (p values). The Wald test *P* values from the subset of genes that pass an independent filtering step, are adjusted for multiple testing using the procedure of Benjamini and Hochberg (padj values). **I** In vitro osteoblastic differentiation of sorted OPCs evaluated at day 21 post-culture by Alizarin Red S coloration. The images are representative of 3 independent cultures. The quantification (means ± SEM) from three independent experiments with 6 mice in total per group is shown. **L** In vitro expanded osteogenic cells from bone fractions were cultured with WT CD11b+ osteoclast progenitors and stimulated with PGE2/Vitamin D3 (VitD3)/Dexamethasone (Dex) for 8 days. OCLs (TRAP-positive) were identified (left, representative images, bars: 50 μm) and quantified (right). Data (means ± SEM) are from 2 independent experiments with 6 mice in total per group. Statistics were calculated with the nonparametric Kruskal–Wallis H test (#$p = 0.033$) and the unpaired two-tailed Student's t test (+/1013 vs WT *$p = 0.02$ and 1013/1013 vs WT *$p = 0.014$). **M** The relative expression levels of osteoclastogenic genes were determined by quantitative PCR in stimulated osteogenic cells (3 mice per group). Each individual sample was run in triplicate and has been standardized for *36B4* expression levels. All mice were littermates, females, and age-matched (8–12 wk-old). Source data are provided as a Source data file.

A slight but significant increase in the frequency of cells in the quiescent G0 state (DAPI^low Ki-67^-) was observed among 1013/1013 SSCs but not in the more differentiated osteoblastic pool (Fig. 5A). The turnover of those cells was then studied by performing a 12-day BrdU pulse-chase assay in vivo (Fig. 5B). Consistent with previous studies[28,69], the fraction of BrdU+ cells in WT SSCs reached ~5%, while we observed a *Cxcr4^1013* allele copy number-dependent reduction in BrdU incorporation within mutant SSCs. No changes were observed among mutant OPCs compared to WT ones. Combined to reduced SSC and OPC numbers in mutant bones (Fig. 2B), these findings are suggestive of reduced cycling and osteogenic differentiation capacities of *Cxcr4^1013*-bearing SSCs.

To gain further mechanistic insights, we investigated the impact of the gain-of-Cxcr4-function on the molecular identity of SSCs by performing RNA-seq analyses of bulk-sorted SSCs from the bone fraction of WT and mutant mice. Biological processes related to cell cycle and osteogenic differentiation were significantly modulated in 1013/1013 SSCs as determined by Gene set enrichment analysis (GSEA) (Fig. 5C). The signature for genes related to cell cycle progression and regulation was reduced in 1013/1013 SSCs (Fig. S3A, S3B). Likewise, genes related to osteogenic differentiation appeared to be decreased in mutant SSCs (Fig. 5D, E). In contrast, key genes involved in both adipogenesis and chondrogenesis were not differentially expressed in mutant SSCs (Fig. S3C). These results were confirmed by microfluidic-based multiplex gene expression analyses (Figs. 5F, G and S3D), thus suggesting that proper Cxcr4 signaling is required for regulating osteogenic specification of SSCs. No changes in expression levels of pro-osteoclastogenic genes were detected in mutant SSCs compared to WT ones (Fig. 5H–J). Therefore, these results unravel a Cxcr4-mediated transcriptional signature in *Cxcr4^1013*-bearing SSCs suggestive of impaired cell cycle progression and defective osteogenic specification.

In adult BM, the majority of OBLs derives from OPCs identified by markers such as osterix (Osx)[3,9,17,28,70–72]. They are predominantly found close to the growth plate cartilage along trabecular bone of the primary spongiosa, and along the metaphyseal cortical bone[70,71,73]. We thus examined whether the gain-of-Cxcr4-function mutation alters the number of Osx-positive OPCs by immunodetection on bone sections. We found fewer Osx-positive OPCs in mutant bones compared to WT (Fig. 5K, L). This decrease was confirmed by flow cytometry in the flushed stromal marrow fraction that encompasses Sca-1-negative and PDGFRα-positive early OPCs with multipotent adipo/osteogenic potential (Fig. 5M)[28,45,70]. Together, these data suggest that the decrease in early and committed OPCs in mutant mice may arise from a defect in osteogenic specification of BM-residing SSCs.

## Cxcr4 desensitization intrinsically regulates in vitro the osteogenic differentiation of skeletal stromal cells

To assess whether Cxcr4 desensitization could regulate SSC fate toward the osteogenic lineage in a cell-intrinsic manner, we first compared in vitro the clonogenic capacities of WT and mutant total skeletal cells. There was a significant decrease in the number of colony-forming units-fibroblast (CFU-Fs) in mutant bone cell cultures that followed a *Cxcr4^1013* allele copy number-dependent pattern (Fig. 6A). These results suggested that impaired Cxcr4 signaling might affect in vitro overall SSC numbers as well as their proliferation. To test this, we evaluated by flow cytometry cell cycle and proliferation of SSCs expanded in vitro using BrdU, Cell Trace Violet (CTV), and DAPI/Ki-67 staining. By day 5 after BrdU pulse, we observed a *Cxcr4^1013* allele dose-dependent reduction in BrdU incorporation within mutant SSCs as compared to WT (Fig. 6B, left panel). Consistently, the fraction of proliferating CTV^low cells was reduced among mutant SSCs three days after loading (Fig. 6B, right panel). This altered proliferative capacity of *Cxcr4^1013*-bearing SSCs was associated with a slight but significant increase in proportions of SSCs in the quiescent G0 state (DAPI^low Ki-67^-), whereas no changes in apoptosis level were observed (Fig. 6C). This might account for the increased doubling time of mutant SSCs as well as their overall reduced number during the culture (Fig. 6D). Altogether, these findings suggest that Cxcr4 desensitization is required in vitro for appropriate SSC proliferation, expansion, and likely maintenance.

Next, we investigated in vitro the osteogenic potential capacities of *Cxcr4^1013*-bearing SSCs[62,63]. Staining of in vitro differentiated osteogenic cells and mineralization capacities by Alp and AR was significantly reduced in cultures from mutant SSCs in an allele dose-dependent manner (Fig. 6E, F, upper panels). Real-time PCR analysis revealed decreased expression of genes encoding osteogenic regulators in mutant cultures (Fig. 6G, upper panels). This was more

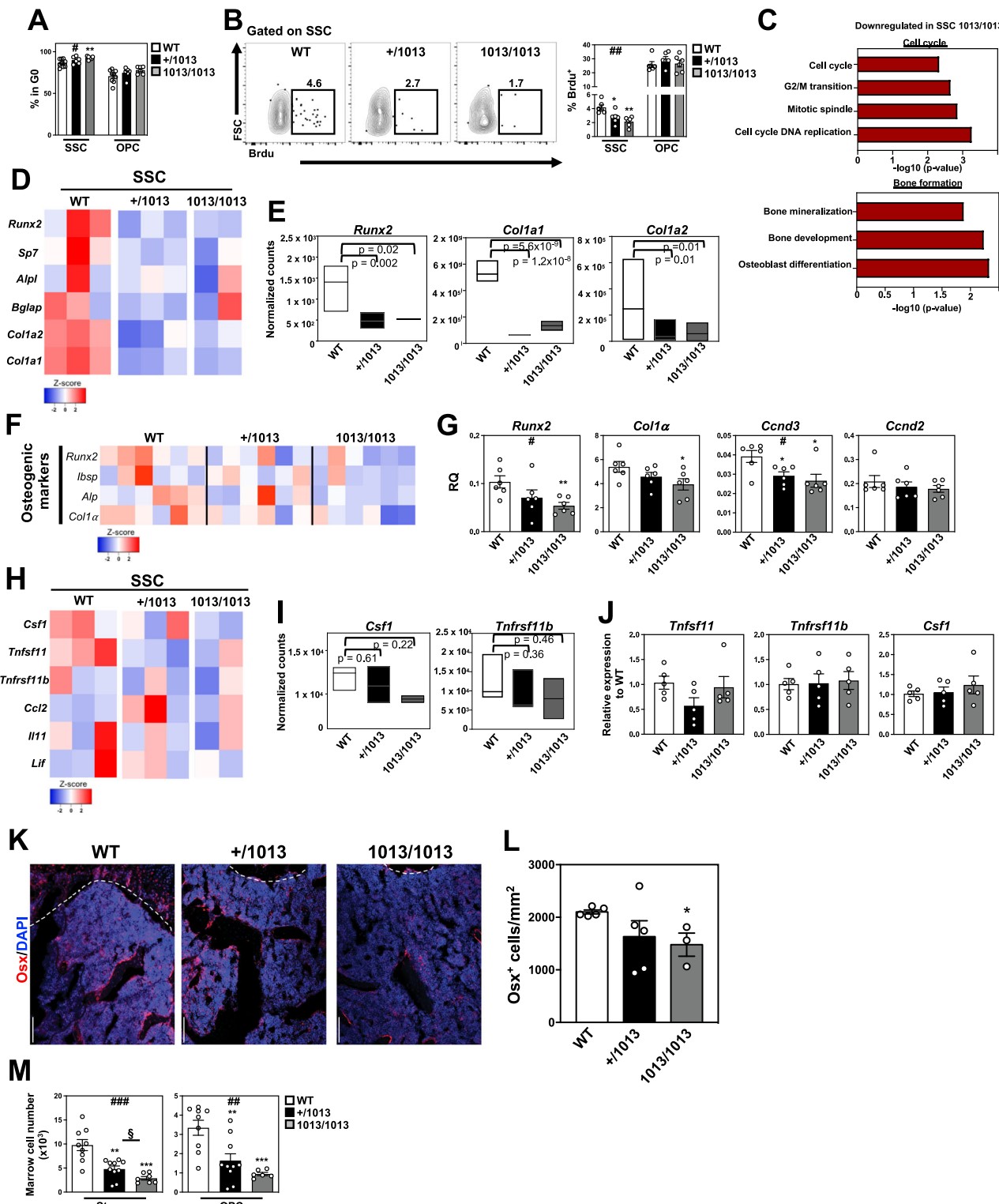

marked for early osteogenic genes downstream the master regulator Runx2 such as *Osx* and particularly evident in the culture of 1013/1013 SSCs, thus suggesting defects at very early stages in the osteogenic differentiation process. In line with this, higher mRNA expression of *Sca1* and *Pdgfrα* was observed in mutant cultures three weeks after pro-osteogenic condition initiation (Fig. S4A, left panel). Consistent with the results obtained with Alcian blue and Perilipin staining on bone sections (Fig. 1G, H), *Cxcr4^{1013}*-bearing SSCs differentiated into adipocytes or chondrocytes similarly to WT SSCs when cultured in vitro with adipogenic or chondrogenic media, respectively (Fig. S4B, S4C).

Collectively, these data reveal in vitro a selective reduction of the osteogenic differentiation capacity of mutant SSCs, and further confirm a pivotal role for Cxcr4 desensitization in regulating this process at very early stages.

**Normalization of Cxcr4 signaling rescues the osteogenic properties of *Cxcr4^{1013}*-bearing mouse skeletal cells**

We then determined whether targeting Cxcr4 signaling would counteract the defective osteogenic fate of mutant SSCs. First, we assessed in vitro the impact of adding AMD3100 every 2 days on the osteogenic

**Fig. 5 | Impaired osteogenic specification of *Cxcr4^1013*-bearing skeletal stromal/ stem cells. A** Ki-67 and DAPI co-staining to analyze by flow cytometry the cell cycle status of SSCs and OPCs from bone fractions. Bar graphs show the percentage of cells (DAPI^low Ki-67^-) in the quiescent G0 phase. Data (means ± SEM) are from three independent experiments with *n* = 9, 6, and 6 mice in total for WT, +/1013, and 1013/ 1013 groups, respectively. Statistics were calculated with the nonparametric Kruskal–Wallis H test (^#*p* = 0.029) and the unpaired two-tailed Student's t test (1013/ 1013 vs WT **p* = 0.0091). **B** Flow-cytometric detection of BrdU staining in SSCs (left). Percentages of BrdU^+ bone SSCs and OPCs after a 12-day labeling period (right). Data (means ± SEM) are from three independent experiments with six mice in total per group. Statistics were calculated with the nonparametric Kruskal–Wallis H test (^##*p* = 0.0021) and the unpaired two-tailed Student's t test (+/1013 vs WT **p* = 0.016, 1013/1013 vs WT **p* = 0.0011). **C** Characterization of some biological processes displaying differential gene expression signatures in sorted SSCs as defined by GSEA and obtained by RNA-seq on 2 (1013/1013) or 3 (WT and +/1013) biological replicates per group with one replicate representing the pool of 3 mice. For significance testing, DESeq2 uses a Wald test (*p* values). The Wald test *P* values from the subset of genes that pass an independent filtering step, are adjusted for multiple testing using the procedure of Benjamini and Hochberg (padj values). **D** RNA-seq-based heatmap representing the relative expression levels of osteo-genic genes. **E** Normalized counts of selected osteogenic genes using the DESeq2 method. Data are represented as floating bars (min to max and line equal median) of the 2 or 3 biological replicates per group. For significance testing, DESeq2 uses a Wald test (*p* values). **F** The heatmap shows the relative expression levels (RQ) normalized for *β-actin* expression levels in each sample of selected genes involved in SSC differentiation towards the osteogenic lineage (6 pools of 100 cells *per* condition) by quantitative PCR. **G** RQ of the most regulated genes involved in differentiation and cell cycle of SSCs. Data (means ± SEM) are from two independent experiments with 6 mice in total per group. Statistics were calculated with the nonparametric Kruskal–Wallis H test (^#*p* = 0.028 for *Runx2*; ^#*p* = 0.011 for *Ccnd3*) and the unpaired two-tailed Student's t test (1013/1013 vs WT **p* = 0.0063 for *Runx2*; 1013/1013 vs WT **p* = 0.048 for *Col1α*; +/1013 vs WT **p* = 0.022 and 1013/1013 vs WT **p* = 0.02 for *Ccnd3*). **H** RNA-seq-based heatmap representing the relative expression levels of osteoclastogenic genes expressed by sorted SSCs. **I** Normalized counts of selected osteoclastogenic genes using the DESeq2 method. Data are represented as floating bars (min to max and line equal median) of the 2 or 3 biological replicates per group. For significance testing, DESeq2 uses a Wald test (*p* values). **J** Relative expression of osteoclastogenic genes in SSCs by quantitative PCR. Each individual sample was run in triplicate and has been standardized for *β-actin* expression levels and presented as relative expression to WT. Data (means ± SEM) are from two independent experiments with 5 mice in total per group. **K** Immunofluorescence showing in red Osterix (Osx)-positive cells and in blue DAPI-stained nuclei in WT and mutant mice femurs (bars: 100 μm). Dashed lines indicate the limit between the cartilage growth plate (above the line) and the bone (below the line). Images are representative of at least 3 independent determinations. **L** Quantification of Osx^+ cells per mm² below the growth plate. Data (means ± SEM) are from 5, 5, and 3 independent mice in total for WT, +/1013, and 1013/1013 groups, respectively. Statistics were calculated with the unpaired two-tailed Student's t test (1013/1013 vs WT **p* = 0.0101). **M** Absolute numbers of the indicated stroma cell subsets from marrow fractions determined by flow cyto-metry. Data (means ± SEM) are from four independent experiments with *n* = 9, 10, and 7 mice in total for WT, +/1013, and 1013/1013 groups, respectively. Statistics were calculated with the nonparametric Kruskal–Wallis H test (^###*p* = 0.0004 for stroma; ^##*p* = 0.0013 for OPC) and the unpaired two-tailed Student's t test (+/1013 vs WT **p* = 0.0011, 1013/1013 vs WT ***p* = 0.0002, +/1013 vs 1013/1013 ^§*p* = 0.033, for stroma; +/1013 vs WT **p* = 0.0049, 1013/1013 vs WT ***p* = 0.0003, for OPC). All mice were littermates, females and age-matched (8–12 wk-old). Source data are provided as a Source data file.

capacities of SSCs. AMD3100-mediated inhibition of Cxcr4 signaling in WT SSCs led to slight changes including decreased numbers of osteogenic cells (Fig. 6E, F, lower panels). By contrast, mutant cultures were highly sensitive to AMD3100 treatment as shown by the nor-malization of Alp and AR colorations 14 and 21 days after differentia-tion, respectively (Fig. 6E, F), and of the expression of *Sca1* and *Pdgfrα* (Fig. S4A). Moreover, AMD3100-mediated reversion of defective osteogenesis within *Cxcr4^1013*-bearing SSC cultures was associated with normalized gene expression of osteogenic master regulators (Fig. 6G, lower panels), thus unraveling that Cxcr4 desensitization intrinsically regulates in vitro the osteogenic differentiation of SSCs.

Then, we assessed the impact of daily intraperitoneal injections for 3 weeks of 5 mg/kg AMD3100 on the bone landscape in adult WT and mutant mice (Fig. 7A). Cxcr4 inhibition decreased slightly the number of WT skeletal cells, and notably OPCs, in the bone fraction (Fig. 7B). In line with this, Opn-stained femoral sections revealed minor alterations in the architecture of WT mice trabecular microstructures upon treatment (Fig. 7C). This was extended to lumbar spine that displayed roughly normal BMD values in treated vs untreated WT mice (Fig. 7D). In 1013/1013 mice, chronic AMD3100 treatment reversed the quantitative defect in skeletal cells by normalizing the numbers of SSCs and OPCs (Fig. 7B). This was not evidenced in +/1013 mice nor associated with a rescue of the trabecular or the cortical network (Fig. 7C, E). However, AMD3100 treatment ameliorated slightly but significantly BMD values of lumbar spine in mutant mice (Fig. 7D), suggesting a correcting effect of Cxcr4-dependent signaling dam-pening on other cell types and mechanisms such as OCLs or LepR-positive BM SSCs as recently reported[31]. Therefore, these data suggest that integrity of Cxcr4 signaling is required for maintaining the osteogenic properties of skeletal cells.

**BM stromal cells from WS patients displayed in vitro impaired osteogenic capacities**

Finally, we sought to investigate if CXCR4 desensitization was mechanistically involved in regulating in vitro the multilineage differ-entiation capacities of human primary BMSCs that constitute a heterogeneous population containing skeletal progenitors[18]. To this end, we analyzed BM samples from two unrelated patients with WS and carrying the heterozygous *CXCR4^R334X* mutation. In parallel, we expan-ded in vitro BMSCs from BM aspirates of seven independent healthy donors. All culture-expanded BMSCs were negative for the CD45 hematopoietic marker lineage but positive for CD73, CD90 and CD105, a combination of markers that are indicative of stromal/fibroblastic cells (Fig. S5A). Both healthy and WS BMSCs were spindle-shaped and fibroblast-like cells and had the ability to form stromal colonies as shown by CFU-F assay (Fig. S5B, S5C). CXCL12 and its two receptors CXCR4 and ACKR3 were readily detectable and found at similar levels between cultured healthy and WS BMSCs (Fig. S5D–F). However, real-time PCR analyses revealed decreased expression of genes encoding early and late osteogenic master regulators in WS BMSC cultures compared to healthy controls (Fig. 8A). In line with this, when equal numbers of cells were plated at the start of the assay, WS BMSCs exhibited defective capacities to generate in vitro osteogenic progeny in contrast to BMSCs harvested from healthy donors (Fig. 8B). In contrast, WS BMSCs were as efficient as control cells to generate adi-pocytes in appropriate culture media condition (Fig. 8C). Therefore, these findings suggest that in vitro osteogenic differentiation of human primary BMSCs requires proper CXCR4 signaling regulation.

## Discussion

In this study, we investigated the regulatory role of CXCR4 signaling termination in the self-renewal and differentiation capacities of adult BM-residing SSCs. We used a knock-in mouse model expressing a naturally occurring WS-linked heterozygous gain-of-function *Cxcr4* mutation as well as human BM samples and clinical data from healthy and WS donors. We demonstrated a mutated allele dose-dependent effect of the WS-linked *Cxcr4^1013* mutation on trabecular bone micro-structures mimicking an osteoporotic-like syndrome, evidenced as well in one-quarter of WS patients. This Cxcr4-mediated reduction in bone content involved both cell-autonomous and cell-extrinsic defects in SSCs (Fig. 8D). Indeed, we provided unanticipated evidence that Cxcr4 desensitization is intrinsically required for regulating in vitro the

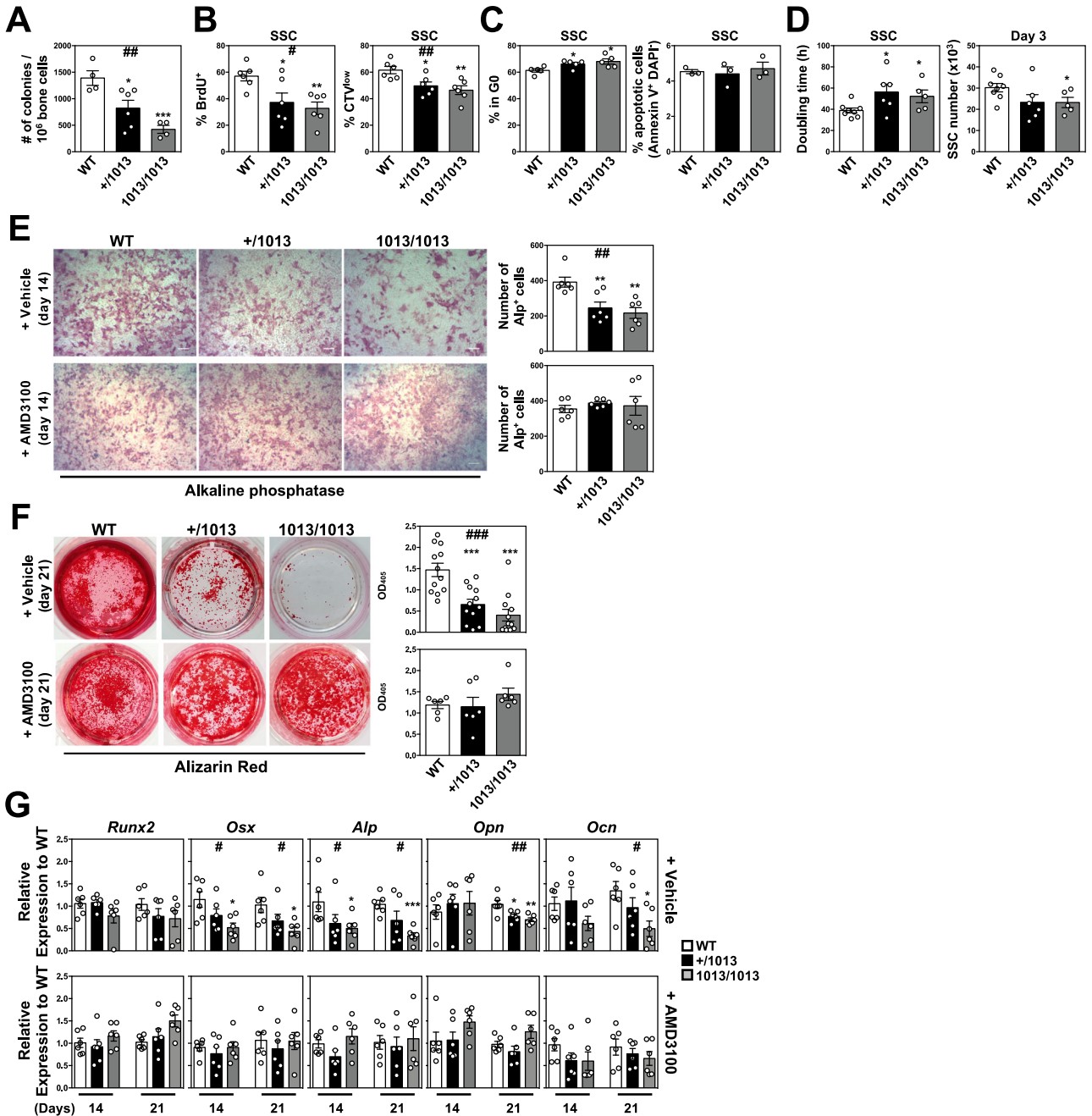

quiescence/cycling balance of SSCs and preserving their osteogenic potential, while it was found to be dispensable for their adipogenic and chondrogenic differentiation. Some other BM cellular components contributed to the dysregulation of the bone phenotype. We observed in trabecular area an increase in OCL number. However, the osteoclastogenic differentiation potential of OCL precursors and the resorptive function of differentiated OCLs were not affected in vitro by the *Cxcr4^1013* mutation. Therefore, the osteopenia might proceed from a deregulated bone matrix resorption that might not be compensated enough by bone-forming cells, enlightening further the strong entanglement between both actors of bone remodeling. Unraveling the mechanisms leading to such an alteration of the skeletal landscape with a focus on the defective osteogenic capacities of SSCs driving the observed excessive bone defect will require further investigations using specific conditional mouse models. Importantly, defective osteogenic capacities were also evidenced in vitro in BMSCs from WS patients. These anomalies establish the C-tail of CXCR4 as an

important regulatory domain of the receptor function in BM stromal cell biology in both mice and humans. In light of previous works[40,41,43,44], our results also suggest that both increased and decreased Cxcr4-mediated signaling negatively impact skeletal stromal elements, thus indicating that fine-tuning of Cxcr4 signaling is critical for maintenance and osteogenic specification of adult SSCs. Although the underlying molecular mechanisms remain to be elucidated, Cxcr4 might act as a rheostat regulating the strength and kinetic of signaling pathways involved in osteogenic fate specification of SSCs. Interestingly, a recent work using a new mouse model of WS further illustrated the crucial role of fine-tuned Cxcr4-mediated signaling in mesenchymal stromal/stem cell (MSC) function[46]. Indeed, the lymphopoiesis process was reduced, as observed in our model[53], because of a dysregulated transcriptome in MSCs isolated from the flushed marrow fraction and characterized by a switch from an adipogenic to an osteolineage-prone program with limited lymphopoietic activity. This might proceed from reduced expression of IL-7 and excessive

**Fig. 6 | Cxcr4 desensitization intrinsically regulates in vitro the osteogenic differentiation of skeletal stromal cells. A** Number of colonies formed from bone fractions in CFU-F assays. Data (means ± SEM) are from two independent experiments with $n = 4$, 6, and 4 mice in total for WT, +/1013, and 1013/1013 groups, respectively. Statistics were calculated with the nonparametric Kruskal–Wallis H test (##$p = 0.002$) and the unpaired two-tailed Student's t test (+/1013 vs WT *$p = 0.029$, 1013/1013 vs WT ***$p = 0.0008$). **B** After in vitro loading with BrdU (5 days) or CTV (3 days), the percentages of BrdU+ (left) or CTVlow (right) cells within WT and mutant bone-derived SSCs were determined by flow cytometry. Data (means ± SEM) are from 3 independent experiments with 6 mice in total per group. Statistics were calculated with the nonparametric Kruskal–Wallis H test (#$p = 0.0231$ and ##$p = 0.0047$ for BrdU+ and CTVlow, respectively) and the unpaired two-tailed Student's t test (+/1013 vs WT *$p = 0.031$, 1013/1013 vs WT **$p = 0.0023$ for BrdU+; +/1013 vs WT *$p = 0.015$, 1013/1013 vs WT **$p = 0.0054$ for CTVlow). **C** Bar graphs show the percentages of cultured WT or mutant SSCs in the quiescent G0 phase (DAPIlowKi-67−, left) or with an apoptotic phenotype (Annexin V+ DAPI−, right) as determined by flow cytometry. Data (means ± SEM) are from three (right panel) or five (left panel) independent SSC cultures per genotype. Statistics were calculated using the unpaired two-tailed Student's t test (+/1013 vs WT *$p = 0.032$, 1013/1013 vs WT *$p = 0.021$). **D** Doubling time (left) and absolute numbers (right) of WT and mutant SSCs after 3 days of culture. Data (means ± SEM) are from 8, 6, and 5 independent SSC cultures for WT, +/1013, and 1013/1013 groups, respectively. Statistics were calculated using the unpaired two-tailed Student's t test (+/1013 vs WT *$p = 0.028$, 1013/1013 vs WT *$p = 0.033$ for doubling-time; 1013/1013 vs WT *$p = 0.048$ for SSC). **E** Alkaline phosphatase (Alp) staining was performed 14 days after initiation of the culture of WT and mutant SSCs in osteogenic medium

supplemented every two days with 10 μM AMD3100 or vehicle (PBS) (bars: 100 μm). Quantitative analyses (number of Alp+ cells) were performed under an inverted microscope. Data (means ± SEM) are from 6 independent cultures per genotype. Statistics were calculated with the nonparametric Kruskal–Wallis H test (##$p = 0.0022$) and the unpaired two-tailed Student's t test (+/1013 vs WT **$p = 0.0075$, 1013/1013 vs WT **$p = 0.0018$). **F** Alizarin Red staining was performed 21 days after initiation of the culture. Quantitative analyses (means ± SEM) of staining were performed using the osteogenesis assay kit in 12 (vehicle) or 6 (AMD3100) independent cultures per genotype. Statistics were calculated with the nonparametric Kruskal–Wallis H test (###$p = 0.0002$) and the unpaired two-tailed Student's t test (+/1013 vs WT ***$p = 0.0005$, 1013/1013 vs WT ***$p < 0.0001$). **G** Expression levels of osteogenic genes were determined by quantitative PCR in 6 independent WT and mutant SSC cultures 14 and 21 days after initiation of the osteogenic culture in the presence or absence of AMD3100. Each individual sample was run in triplicate and was standardized for *β-actin* expression levels. Results (means ± SEM) are expressed as relative expression compared to WT samples. Statistics were calculated with the nonparametric Kruskal–Wallis H test (#$p = 0.038$ and 0.0205 for Osx days 14 and 21, respectively; #$p = 0.024$ and 0.015 for Alp days 14 and 21, respectively; ##$p = 0.0063$ for Opn days 21; #$p = 0.026$ for Ocn days 21) and the unpaired two-tailed Student's t test (1013/1013 vs WT *$p = 0.0107$ and 0.013 for Osx days 14 and 21, respectively; 1013/1013 vs WT *$p = 0.035$ and ***$p = 0.0001$ for Alp days 14 and 21, respectively; +/1013 vs WT *$p = 0.022$ and 1013/1013 vs WT **$p = 0.0056$ for Opn days 21; 1013/1013 vs WT **$p = 0.0108$ for Ocn days 21). All mice were littermates, females, and age-matched (8–12 wk-old). Source data are provided as a Source data file.

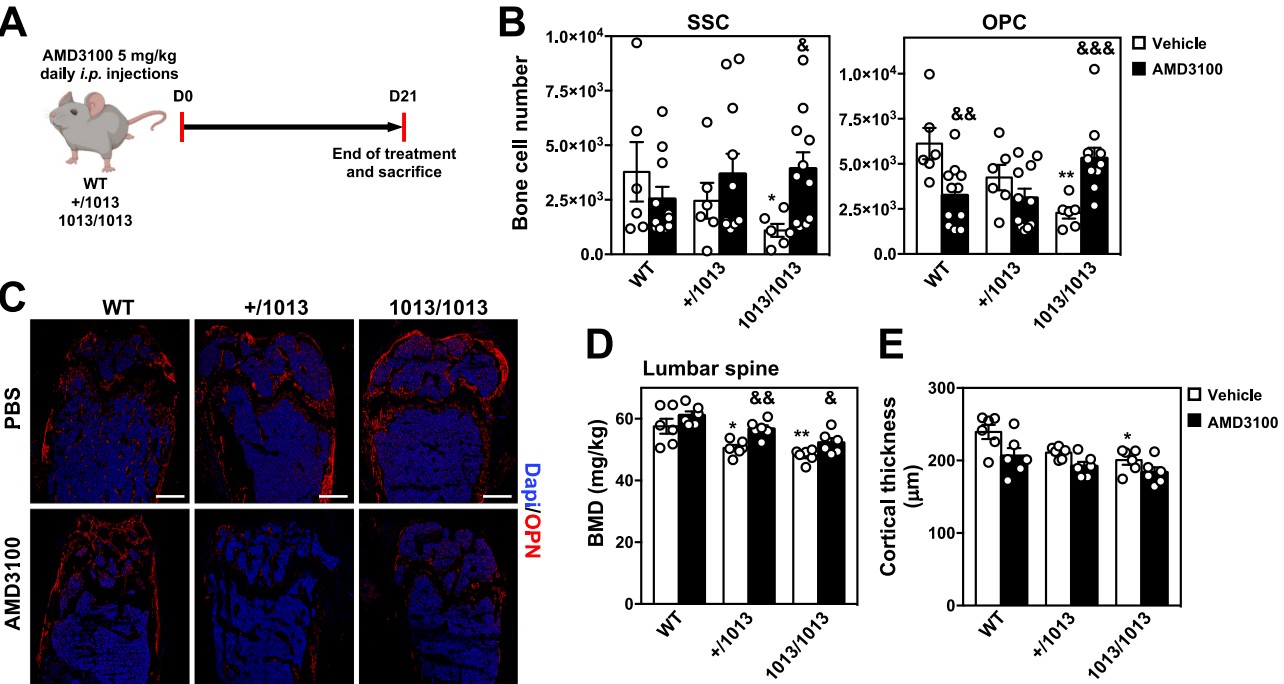

**Fig. 7 | Normalization of Cxcr4 signaling rescues the osteogenic properties of *Cxcr4^1013*-bearing mouse skeletal cells. A** Schematic diagram for daily AMD3100 intra-peritoneal (*i.p.*) injection for 21 days in WT and mutant mice. Mouse icons were created using the Biorender software (Biorender.com, agreement number: VX255VH9TZ). **B** Absolute numbers of the indicated stroma cell subsets from bone fractions of WT and mutant mice determined by flow cytometry. Data (means ± SEM) are from 2 independent experiments with 6 PBS injected mice and 11 AMD3100-injected mice in total *per* genotype. Statistics were calculated using the nonparametric Mann–Whitney test, two-sided, 1013/1013 vs WT *$p = 0.041$, 1013/1013 (AMD) vs 1013/1013 (vehicle) &$p = 0.01$ for SSC; 1013/1013 vs WT **$p = 0.0022$, WT (AMD) vs WT (vehicle) &&$p = 0.0065$, 1013/1013 (AMD) vs 1013/1013 (vehicle) &&&$p = 0.0003$ for OPC. **C** BM sections from WT and mutant mice treated with vehicle (PBS) or AMD3100 were immunostained for Opn in association with DAPI.

Bars: 500 μm. Images are representative of 3 independent determinations. **D** Bone mineral density (BMD) values of lumbar spine from 6 treated mice in total per group are shown. Statistics were calculated using the nonparametric Mann–Whitney test, two-sided, +/1013 vs WT *$p = 0.041$, 1013/1013 vs WT **$p = 0.0022$, +/1013 (AMD) vs +/1013 (vehicle) &&$p = 0.0087$, 1013/1013 (AMD) vs 1013/1013 (vehicle) &$p = 0.026$. **E** Cortical thickness was measured in sections stained with Toluidine Blue. Both cortices were measured and data are presented as a mean of both cortices from 6 mice in total per group. Statistics were calculated using the nonparametric Mann–Whitney test, two-sided, 1013/1013 vs WT **$p = 0.026$. Data (means ± SEM) displayed in (**D**) and (**E**) are from 2 independent experiments. All mice were littermates, females and age-matched (8–12 wk-old). Source data are provided as a Source data file.

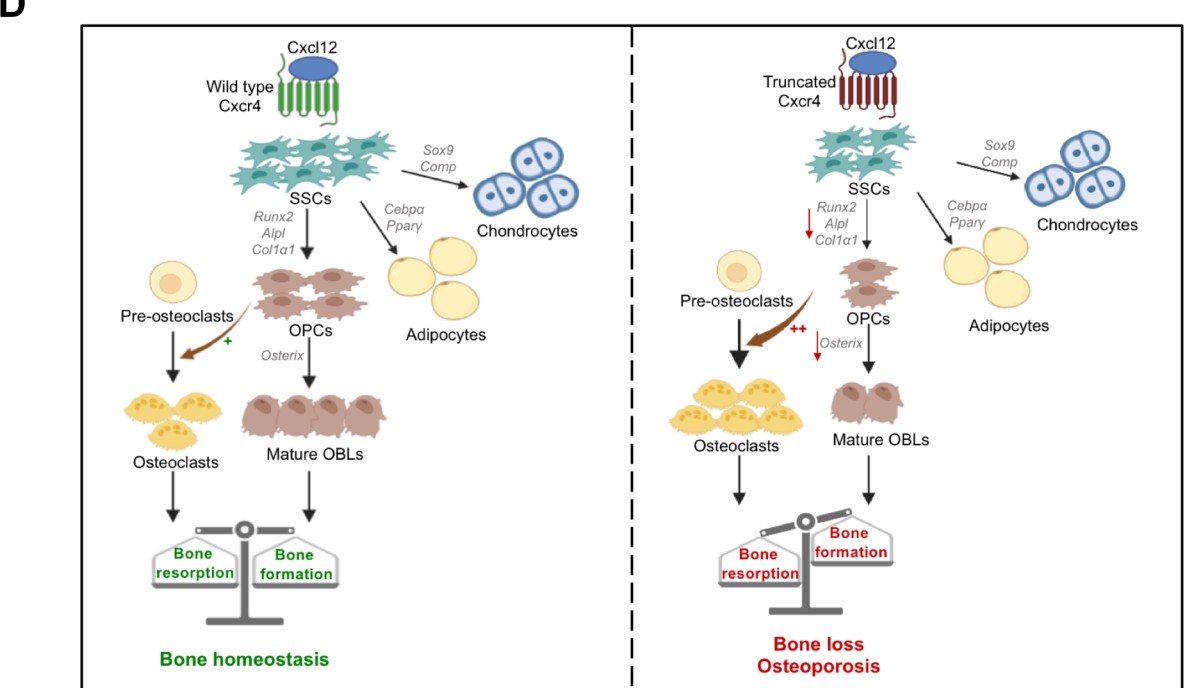

signaling through the lymphotoxin beta receptor in MSCs. Whether these changes are observed in the BM of our mouse model and how they would contribute to the bone loss warrant further investigations. Moreover, mice deficient for the gene encoding the transcription factor Ebf3 display an opposite BM phenotype to the one of *Cxcr4[1013]*-bearing mice, characterized by osteosclerosis with HSC depletion and reduced expression of niche factors[69]. This was related to the uncontrolled ability of *Ebf3*-deficient SSCs to differentiate into OBLs. Further

studies are required to address the status of Ebf3 and downstream target genes that act to modulate osteogenic fate of SSCs in *Cxcr4[1013]*-bearing mice. A potential crosstalk between distinct SSC subsets, either prone to differentiate into osteochondro-lineage cells or perivascular and adipocyte lineage cells, has been reported[74]. This seems to imply ligand-receptor gene pairs such as TGFβ, WNT or BMP ligands and their cognate receptors that regulate SSC fate decision. Whether and how the Cxcl12/Cxcr4 signaling axis contributes to

**Fig. 8 | BM stromal cells from WS patients displayed in vitro impaired osteogenic capacities. A** Relative expression levels of osteogenic genes were determined by quantitative PCR at day 14 in osteogenic-induced cultures of two WS patients-derived BMSCs and 7 healthy donors-derived BMSCs. Each individual sample was run in triplicate and was standardized for *36B4* expression levels. Results (means ± SEM) are expressed as relative expression compared to healthy samples (set at 1 and representing the mean of the 7 healthy donors) and are from 2 independent experiments. Statistics were calculated using the nonparametric Mann–Whitney test, two-sided, **$p = 0.0079$ for OSX, RUNX2, and OCN; *$p = 0.0179$ for OPN. **B** Alizarin Red staining was performed 21 days after initiation of the culture of $1.5 \times 10^3$ healthy or WS BMSCs in pro-osteogenic medium (left panel). Representative images for healthy (H) and WS donors #1 and #2 are shown. Quantitative analyses of staining (means ± SEM) were performed using the osteogenesis assay kit

in 3 independent cultures with 2 WS and 6 healthy donors (right panel). Statistics were calculated using the nonparametric Mann–Whitney test, two-sided, **$p = 0.0022$ for healthy. **C** Oil Red O staining was performed 21 days after initiation of cultures of healthy or WS BMSCs in pro-adipogenic differentiation medium. Bars: 200 μm. **D** Proposed model: Proper Cxcr4 signaling termination is essential for bone tissue homeostasis. Absence of Cxcr4 desensitization leads to imbalance in bone remodeling with decreased OBL-mediated bone formation and increased OCL-mediated bone resorption, leading to severe trabecular and cortical alterations and a subsequent osteoporotic-like phenotype. Mechanistically, impaired Cxcr4 desensitization disrupts the osteogenic commitment of SSCs, while strikingly increasing their pro-osteoclastogenic capacities. Graphical abstract was created using the Biorender software (Biorender.com, agreement number: EL255UV5RI). Source data are provided as a Source data file.

these regulatory mechanisms across SSC types remains also to be explored.

We reported that loss of Cxcr4 signaling termination impairs overall number, impedes cell cycle progression, and limits osteogenic differentiation of SSCs. Indeed, mutant mice have a global alteration of the bone stromal landscape, including decreased numbers of SSCs and their progeny including early and committed OPCs and impaired architecture of trabecular and cortical bone microstructures that occurred in a mutated allele copy number-dependent manner. Altogether, these findings indicate that the gain-of-*Cxcr4*-function mutation promotes a reduced OBL commitment and differentiation, but does not alter the bone forming activity of individual OBL. Congruent with this, chronic treatment with AMD3100 normalized in vitro the osteogenic properties of mutant SSCs. Impaired Cxcr4 desensitization might alter the balance between quiescence and differentiation of mutant SSCs and reduce the number of osteogenic-endowed precursors. Currently, the prevailing view is that BM Cxcl12-expressing stromal cells display slow cell cycle progression and constitute an active source of trabecular and cortical OBLs under physiological conditions, as well as in response to injury[6,9,28,31,69]. We found a higher proportion of quiescent SSCs in mutant mice that was particularly evident in 1013/1013 SSCs, thus suggesting the importance of Cxcr4 desensitization in controlling SSC proliferation and quiescence and likely their capacity to give rise to osteogenic cells.

Loss of bone content in mutant mice was accompanied by a higher number of OCLs within the trabecular bones, possibly reflecting that the *Cxcr4* mutation was intrinsically perturbing the OCL differentiation process. This seems not to be the case since we showed that defective Cxcr4 desensitization did not increase in vitro differentiation of OCLs from mutant BM progenitors, nor their mineral matrix resorbing capacities. BM chimeras leading to a WT hematopoietic development into a mutant bone environment further ruled out the sole involvement of an uncontrolled bone resorption due to defective OCLs with excessive activity. These cells derive from monocytic lineage precursors upon stimulation by RankL and M-Csf[75,76]. In a constant cross interaction between the bone forming and the bone resorbing pathways, these osteoclastic cytokines are produced by mature and immature stromal cell populations within the BM[9,77,78]. While we did not observe increased *Rank-L*, *M-Csf*, or *Opg* (encoding a Rank-L antagonist) gene expression in sorted committed OPCs from mutant bones, we cannot exclude that modification of the bone stroma due to osteogenic defects might in turn modulate the production of osteoclastic factors from the mutant bone environment. Supporting this assumption, co-cultures of WT OCL progenitors with in vitro-expanded *Cxcr4^1013*-bearing bone-derived stromal cells, stimulated to produce osteoclastogenic factors, led to enhanced osteoclastogenesis compared to WT stromal cells. No increase in *Rank-L* or *M-Csf* expression nor decrease in *Opg* level could be detected, suggesting the involvement of other additional mediators. It has recently been shown that intrinsic aging of SSCs resulted in higher proportion of stromal lineages producing pro-inflammatory and pro-resorptive factors,

promoting myeloid skewing, and osteoclastic activity[30]. Whether a similar mechanism occurs in *Cxcr4^1013*-bearing mutant mice remains to be characterized. Moreover, BM chimeras leading to a mutant hematopoietic development into a WT bone environment displayed dysregulated bone landscape, thus emphasizing that transplanted hematopoietic cells can participate in bone loss through direct and/or indirect actions on osteoclastogenesis. Despite altered hematopoiesis in such chimeric mice, the myeloid skewing reported elsewhere[53] might account for excessive OCL number. As well, the reduced lymphopoiesis observed[46,53] still led to generation of mature B and T lymphocytes that are present in the BM and hence may act as potential actors in bone erosion as they can produce osteoclastogenic mediators such as Rank-L in non-physiological settings[79]. Whether such action of mutant hematopoietic donor cells recreates a pro-osteoclastogenic environment through a remodeling of the myeloid and lymphoid compartments deserves further investigations. Interestingly, the dysregulated bone landscape in BM chimeras with a mutant hematopoietic compartment in WT recipients was even stronger than that observed in mutant mice at steady-state. This observation raises the intriguing possibility of a pro-osteogenic effect of one or several *Cxcr4^1013*-bearing radioresistant cell type(s). To test this possibility, further work will be required using mouse model where the mutation is carried in a cell-specific manner.

Osteogenesis is regulated, among different mechanisms, by undifferentiated skeletal cells and more specified osteolineage cells that express factors promoting or preventing their own differentiation into OBLs[23,59,69,80]. In BM, HSPC niches constitute critical spatiotemporal regulatory units composed of multiple cell populations of hematopoietic and non-hematopoietic origin cross-interacting with each other's in a dynamic setting[1,3,9,81,82]. This implies that immune and vascular cells among others may influence the osteogenic differentiation process[83]. Again, in BM chimeras in which *Cxcr4^1013*-bearing HSPCs were differentiating into a WT bone environment, we reported a similar bone loss as observed in mutant mice, thus indicating cell-extrinsic Cxcr4-mediated regulation of the skeletal landscape. This also suggests that neither the epiphyseal cartilage nor any developmental defect contribute to impaired trabecular bone architecture in adult mutant mice, and further supports the notion that HSPCs, as osteolineage cells do, express regulating osteogenic factors such as BMP-2, BMP-7, and WNT3a, that are particularly involved in SSC osteogenesis specification[17]. Whether and how hematopoietic cells, or other BM components such as vascular cells, participate in the defective osteolineage specification of SSCs in *Cxcr4^1013*-bearing mice deserves further investigations.

Finally, we reported that five out of nineteen patients with WS and carrying distinct autosomal-dominant mutations in *CXCR4* exhibit a decrease in BMD at different anatomical sites. Although this would merit to be extended to a larger cohort before introducing any potential bone-affecting drugs, these data suggest that accelerated osteopenia/osteoporosis and increased risk of fractures may constitute a novel feature of WS. Lack of CXCR4 desensitization could be

mechanistically involved in such anomaly since BMSCs from WS patients carrying a heterozygous *CXCR4* mutation displayed in vitro impaired capacities to differentiate into osteogenic, but not adipogenic, cells. Strikingly, we observed that chondro- and adipo-genic differentiation of murine mutant SSCs was normal both in situ and in vitro. Considering recent studies unraveling human SSCs expressing the CXCL12/CXCR4 axis with osteoblastogenic and, depending on their tissue origin, adipocytic potential[18,84], our findings pave the way for exploring the BM of WS patients in search for potential defect(s) in these skeletal populations.

## Methods

### Healthy and WS donors and bone mineral density measurements

Investigations of human BM samples were performed in compliance with Good Clinical Practices and the Declaration of Helsinki. The study was approved by the Ethical Board Ile-de-France X. Recruited WS patients were not compensated and gave their written informed consent for participating to the clinical study that has been approved by NIAID Institutional Review Board (IRB). Cryopreserved BM aspirates from two WS patient (NIH protocol 09-I-0200) were provided through a NIH Material Transfer Agreement. The samples were anonymized. BM samples from seven healthy donors that were matched for age and sex and used as control subjects were isolated from hip replacement surgery samples (Protocol 17-030, n° ID-RCB: 2017-A01019-44). Primary BMSCs from healthy and WS donors were amplified and used at passage 1 to 3. For BMD assessment, data were collected from nineteen WS patients as part of an IRB approved clinical protocol conducted at the NIH (NIAID Protocol #2014-I-0185, IND # 118767). Patients had a baseline bone density scan as part of a drug treatment trial (NCT02231879) comparing 1 year of twice daily filgrastim (Neupogen) versus plerixafor (Mozobil) in a randomized, blinded crossover design. There were 13 women and 6 men with an average age of 30.5 years (range 10–56). The samples were anonymized. Patients had been on filgrastim (Neupogen) for an average of 5.7 years prior to enrolling in the trial (range 0–27). 6 of the 19 had not used filgrastim regularly prior to trial enrollment. BMD values expressed as T- or Z-scores were measured by total body dual-energy X-ray absorptiometry with a Lunar iDXA densitometer (GE Healthcare). Five WS patients had abnormal screening bone density by WHO criteria, anonymized at the start of the Phase 3 trial (Table 1), while the other 14 patients had normal bone density.

### Mice and genotyping

All mice were bred in our animal facility under a 12 h light/dark cycle, specific pathogen-free conditions (EOPS status) and fed *ad libitum*. For breeding, mice were in conventional cages with filter top. For experimentation, mice were housed in individually ventilated cages. All experiments were performed in accordance with the European Union guide for the care and use of laboratory animals and have been reviewed and approved by institutional review committees (CEEA-26, Animal Care and Use Committee, Villejuif, France and Comité d'Ethique Paris-Nord/N°121, Paris, France). *Cxcr4*<sup>+/1013</sup> (+/1013) mice were generated by a knock-in strategy[51]. Homozygous *Cxcr4*<sup>1013/1013</sup> (1013/1013) mice were obtained by crossing heterozygous +/1013 mice. WT mice were used as controls. Unless specified, all mice were littermates, females, and age-matched (8–12 wk-old). Adult Boy/J (CD45.1) (Charles River) mice were used as BM donors. Daily observation was performed to ensure that no animal was left in a state of pain or suffering during experimentation. Euthanasia was performed by increasing gradient of $CO_2$.

### Sample isolation in mice

Mouse SSCs were obtained from bones after centrifugation of intact femurs, tibias, and hips to flush out the BM cells. Flushed long bones were cut into fine pieces before enzymatic digestion with 2.5 U/mL collagenase type I (Thermofisher) for 45 min at 37 °C under agitation. Released cells were filtered and washed with PBS, 2% FBS (Fetal Bovine Serum). Cell numbers were standardized as total counts per two legs. Peripheral blood was collected by cardiac puncture. Freshly isolated cells were either immunophenotyped, incubated at 37 °C for 60 min in RPMI 20 mM HEPES 0.5% BSA (Euromedex) prior to chemokine receptor internalization studies, or expanded in αMEM medium supplemented with 10% FBS, 1% P/S (penicillin 100 Units/mL, streptomycin 100 Units/mL, Gibco) and 50 μM β-mercaptoethanol (PAN Biotech). For BMD quantification, lumbar spines were fixed overnight in ethanol 70° and analyzed by dual-energy X-ray absorptiometry with an ultra-focus DXA densitometer (Faxitron). Quantifications were made on a ROI of 2 lumbar spines.

### Flow-cytometric analyses

Mouse and human staining analyses were carried out on an LSRII Fortessa flow cytometer (BD Biosciences) using the antibodies (Abs) described in Table S1. A Live/Dead Fixable Aqua Dead Cell Stain Kit (Biolegend) was used. To assess the compartmentalization of CXCR4 and ACKR3, human BMSCs were incubated with saturating concentrations of non-conjugated mouse anti-human CXCR4 or ACKR3 Abs, washed in PBS, fixed and permeabilized using the BD Cytofix/Cytoperm Fixation/Permeabilization Kit (BD Biosciences). BMSCs were subsequently stained with anti-CXCR4 and -ACKR3 conjugated mAbs, or the corresponding isotype control, at 4 °C for 30 min and then analyzed by flow cytometry. FACS Diva software version 7 (BD) were used for collecting data. FLOWJO v10.7 (BD) and GraphPad Prism v8.0e (GraphPad Software Inc.) were used for analyzing flow cytometric data.

### In vitro functional assays

Mouse CFU-Fs were performed by plating $1 \times 10^5$ bone cells at passage 2–3 from WT and mutant mice. Human CFU-Fs were performed by plating $0.2 \times 10^3$ BMSCs into a 25 cm² flask at passage 3 from healthy or WS donors. After 7 or 10 days of culture, colonies were fixed with ethanol 70%, stained with 2% crystal violet (Sigma-Aldrich), and counted with a binocular magnifying glass. For chemotaxis assays, $5 \times 10^4$ SSCs were added to the upper chambers of a 24-well plate with 8-μm-pore-size Transwell inserts (EMD Millipore) containing or not 1 nM Cxcl12 (R&D Systems) in the lower chamber. For inhibiting Cxcr4-mediated signaling, 10 μM AMD3100 (Sigma-Aldrich) was added in the upper and lower chambers. After 24 h, membranes were removed and fixed in 4% paraformaldehyde (PFA). The cells that migrated to the lower side of the membrane were stained with 0.1% crystal violet and three fields from each insert were counted under a light microscope. Cxcr4 and Ackr3 internalization assays were performed by incubating total bone cells at 37 °C for 45 min with 10 nM Cxcl12. Then the reaction was stopped by adding ice-cold RPMI and quick centrifugation at 4 °C. After one wash in acidic glycine buffer at pH = 4.3, levels of Cxcr4 and Ackr3 membrane expression were determined by flow cytometry. Cxcr4 or Ackr3 expression was calculated as follows: (Cxcr4 or Ackr3 geometric MFI of treated cells/Cxcr4 or Ackr3 geometric MFI of unstimulated cells) × 100; 100% corresponds to receptor expression at the surface of cells incubated in medium alone. For the chemokine scavenging assay, cultured SSCs were harvested by trypsinization and placed in complete medium for 90 min at 37 °C and 5% $CO_2$ to normalize receptor expression. $4 \times 10^6$ cells/mL were pre-incubated with 100 μM CCX733, a functional Ackr3 antagonist or vehicle in 1% BSA/PBS for 45 min at room temperature (RT). Then, $2 \times 10^6$ cells/mL were incubated in presence of 5 nM AF647-Cxcl12 (Almac) in 1% BSA/PBS during 45-60 min at 37 °C to allow internalization or on ice to inhibit this process. Cells were washed with 1% BSA/PBS and then either treated with an acidic glycine wash buffer pH = 2.7 for 3 min to dissociate cell-surface-bound chemokine, or washed with PBS to estimate internalized plus cell-surface-bound control. AF647 fluorescence

(geometric MFI) was determined by flow cytometry. Phosphoflow assays were performed with the PerFix EXPOSE kit (Beckman Coulter) on cultured SSCs and an anti-phospho Erk (pT202/pY204) was used. Fold change was calculated as follows: (Phospho-Erk geometric MFI of stimulated cells/Phospho-Erk geometric MFI of unstimulated cells).

## In vivo functional assays

For BM transplantation experiments, $1.5 \times 10^6$ total marrow cells from young CD45.1$^+$ WT mice were injected i.v. into lethally irradiated (two rounds of 5.5 Gy separated by 3 h) young CD45.2$^+$ WT, +/1013, or 1013/1013 recipient mice. For reverse experiments, $1.5 \times 10^6$ total marrow cells from CD45.2$^+$ WT, +/1013, or 1013/1013 mice were injected into lethally irradiated CD45.1$^+$ WT recipient mice. Chimerism was analyzed 3 or 16 weeks after transplantation. For Cxcr4 blockade experiments, mice were daily injected intraperitoneally with 5 mg/kg AMD3100 or PBS during 3 weeks. BM were harvested 2 h after the last injection and analyzed by flow cytometry and imaging.

## ELISA

Supernatants of culture-expanded human BMSCs were analyzed using a standardized ELISA for human CXCL12 (Quantikine; R&D Systems).

## Bone immunostaining and histomorphometry

Mouse bones were fixed in 4% PFA overnight followed by one-week decalcification in EDTA (0.5 M) at pH 7.4 under agitation. Bones were incubated in PBS with 20% sucrose and 2% polyvinylpyrrolidone (PVP) (Sigma) at 4 °C overnight and then embedded in PBS with 20% sucrose, 2% PVP, and 8% gelatin (Sigma) before storage at −80 °C. Sections of 30-µm thick were rehydrated in PBS 1X, incubated 20 min at RT in PBS with 0.3% triton X-100, saturated in blocking solution (PBS with 5% BSA), and finally incubated with primary Abs (Table S2). After washing, secondary Abs were incubated for 1 h at RT with DAPI for nuclear staining and mounting using Permafluor mounting medium (Thermofisher). Images were acquired using TCS SP8 confocal microscope and processed using Fiji software. For alcian blue and perilipin A staining, fixed and decalcified femur bones were embedded in paraffin, sectioned (7-µm thick) and deparaffinized with xylene. Staining of cartilage tissues was performed with a 1% alcian blue solution for 30 min. Images were acquired using a LEICA DM4000B microscope equipped with a DFC425C camera and processed with the Leica Application Suite V3.8 software. For perilipin A staining, heat induced epitope retrieval was performed in citrate sodium buffer solution. Sections were saturated for 1 h in PBS 1% BSA at RT, washed in PBS 0.2% BSA and 0.1% Triton X-100, and incubated with anti-perilipin A Ab in PBS BSA 1% overnight at 4 °C. After washing, sections were incubated with TRITC-coupled rabbit anti-guinea pig Ab in PBS 1% BSA for 45 min and counterstained with DAPI. For Osx staining, 16 µm frozen sections were permeabilized in TBS-0.3% Triton X-100 for 10 min and blocked in TBS-2.5% BSA-2.5% Donkey Serum for 1 h at RT. Sections were incubated with anti-Osx Ab (rabbit, Santa Cruz SC-22536R) in blocking solution overnight at 4 °C. After washing with TBS + 0.025% Triton X-100, sections were incubated in donkey anti-rabbit secondary Ab dylight 550 (SA5-10039, Invitrogen) in blocking solution. After washing, sections were incubated 15 min at RT in DAPI at 0.1 µg/mL prior to mounting in GB-Mount (Diagomics). Image acquisitions were done using the ApoTome optical sectioning system (Zeiss) with an inverted microscope (Zeiss Axio Observer Z1). Osx quantification was performed using the ICY software. For human BMSC immunofluorescence studies, cells were plated on coverslips and fixed with 4% PFA in PBS. Fixed cells were permeabilized with Triton X 0.3% for 10 min, blocked with PBS 5% BSA, 5% goat serum, and incubated with unlabeled primary CXCL12 mAb overnight at 4 °C followed by secondary AF633-coupled goat anti-mouse polyclonal Ab (Invitrogen) and the nuclear dye Hoechst 33342. Images were obtained with a Plan-Apochromatic objective using the LSM800 confocal microscope (Carl Zeiss). Sections were acquired as serial z stacks (0.39 µm apart) and were subjected to three-dimensional reconstruction (Zen 2.3 System).

Bone histomorphometry was performed in plastic samples, allowing the measurements of bone formation and resorption parameters. Mouse femurs were fixed in ethanol 70°, dehydrated, and embedded in methyl methacrylate resin. Five micrometer-thick coronal sections were cut parallel to the long axis of the femur using an SM2500S microtome (Leica, Germany). Sections were deplastified, rehydrated, and stained with toluidine blue or with naphthol 3-hydroxy-2-naphthoic acid 4-chloro-2-methylanilide (ASTR phosphate, Sigma, St Louis, France) for detecting mature osteoclasts with TRAP staining. Quantifications were made on a polarizing microscope (Nikon) using a software package (Bonolab) developed for bone histomorphometry (Microvision, France). To allow the measure of dynamic parameters of bone formation, mice were intraperitoneally injected with tetracycline (20 mg/kg) and calcein (10 mg/kg; Sigma) 5 days and 1 day, respectively, before being killed. Two 12-µm-thick unstained sections were taken for measurement of the dynamic parameters under UV light. The matrix apposition rate (MAR) was measured using the Microvision image analyzer by a semiautomatic method using tetracycline and calcein double-labeled bone surfaces. The mineralizing surfaces (MS/BS) were measured in the same areas using the objective eyepiece Leitz integrate plate II. When specified, the cortical thickness was measured in paraffin-embedded sections stained with Toluidine Blue. The 2 cortices were measured using histomorphometry software and expressed as mean of both cortices for each sample. All the histomorphometric parameters were recorded in compliance with the recommendation of the American Society for Bone and Mineral Research Histomorphometry Nomenclature Committee. Five to six animals per genotype were analyzed by two different investigators.

## Bone structure analysis by micro-computed tomography

Femurs were collected for bone microarchitecture analysis after fixation and before decalcification. They were analyzed with high-resolution microcomputed tomography (micro-CT) using a Skyscan 1272 microCT (SkyScan, Kontich, Belgium). Measurements were made on the femurs using the following acquisition parameters: voltage 60 kV, pixel size 6 µm, Filter Alu + 0.5 mm. After 3-dimensional images reconstruction with NRecon®, analyses were performed on the trabecular and cortical region (1.72 mm and 0.43 mm thickness, respectively). Morphometric parameters such as Bone Volume/Tissue volume (BV/TV, %), Trabecular number (Tb.Nb, 1/mm), Trabecular Separation (Tb.Sp, mm), Cortical Bone Volume (Ct.BV, mm³), and Cortical Thickness (Ct.Th, mm) were assessed.

## Cell culture and differentiation

Mouse osteoblastic differentiation was performed for 3 weeks in α-MEM medium with 10% FBS, 1% P/S, 50 µM β-mercaptoethanol supplemented with 50 µg/mL L-ascorbic acid and 10 mM glycerophosphate (Sigma) either from SSCs or sorted OPCs. Alkaline phosphatase staining was performed after 14 days of differentiation according to the Alkaline phosphatase Kit (Sigma). At day 21, cultures were fixed with 4% PFA, stained with alizarin red, and quantified using the Osteogenesis assay kit (Millipore). When specified, AMD3100 (versus vehicle) was added into the osteogenic medium every 2 days at 10 µM, respectively. Chondro- and adipo-genic differentiations of SSCs were performed according to the StemPro-Chondrogenesis or -Adipogenic Differentiation Kits (ThermoFisher) for 2 weeks. After fixation, cells were treated with either Alcian Blue 1% (Sigma) to stain chondrocyte matrix or Oil Red O solution (Sigma) to reveal lipid droplets. For in vitro human osteogenic differentiation assays, expanded BMSCs were seeded at $3 \times 10^3$ per cm² in α-MEM supplemented with 10% FBS and 1% antibiotics. After cell adhesion, medium was replaced by α-MEM supplemented with 10% FBS, 1% antibiotics and 0.1 µM

dexamethasone, 0.05 mM L-ascorbic acid-2-phosphate, and 10 mM β-glycerophosphate. Medium was changed every 2 days during 3 weeks. Quantification of mineralization was performed after Alizarin Red S staining[85]. Human adipogenic differentiation assays were performed as described for the murine ones.

## Osteoclast differentiation and functional analysis

OCLs were differentiated in vitro as described[86]. Briefly, $2.3 \times 10^5$ BM cells/cm² were plated in MEM-alpha (ThermoFisher) complemented with 5% serum (Hyclone, GE Healthcare), 1% P/S, 50 μM 2-mercaptoethanol, 25 ng/ml M-Csf and 30 ng/ml Rank-L (R&D Systems). OCL differentiation (multinucleated TRAP⁺ cells) was quantified at day 5 after TRAP coloration using the leukocyte acid phosphatase kit (Sigma). Matrix dissolution activity was evaluated by seeding a total of $2 \times 10^4$ differentiated OCLs on 96-well osteoassay plates (Corning) in α-MEM containing 10% FBS and 30 ng/ml Rank-L. After 3 days, medium was removed and cells were detached by the addition of water. Resorbed areas were quantified using Fiji/ImageJ software[87].

## Co-culture assays between osteogenic cells and osteoclast precursors

Mouse osteogenic cells were isolated from the bone fraction of WT or mutant mice as indicated above. They were expanded in α-MEM medium with 10% FBS, 1% P/S, 50 μM β-mercaptoethanol for 2 weeks, passaged and plated in 96-well plates ($2 \times 10^4$ cells/well). Osteoclast precursors were obtained from CD11b⁺-enriched cells (with CD11b-Microbeads, Miltenyi Biotec, France) from the flushed marrow fraction of WT mice as previously described[86]. They were added, in co-culture, to WT or mutant expanded osteogenic cells ($5 \times 10^4$ cell/well) and stimulated with 1,25-dihydroxy vitamin D3 (vitD3, 10 nM), prostaglandin E2 (PGE2, 1 μM) and Dexamethasone (Dex, 50 nM) as previously described[65] in order to stimulate osteogenic cells to produce RankL/MCSF and to inhibit OPG production[88,89]. OCL differentiation (multinucleated TRAP⁺ cells) was quantified at day 8 after TRAP coloration as indicated above.

## Cell cycle, viability, survival, and proliferation assays

For flow cytometry-based cell cycle analyses, bone cells were permeabilized, fixed with the FOXP3 permeabilization kit (Foxp3/Transcription Factor Staining Buffer Set; eBioscience), and labeled with a Ki67 Ab and DAPI. For BrdU assays, mice were injected intraperitoneally with 180 μg BrdU (Sigma) and maintained with drinking water containing 800 μg/ml BrdU and 1% glucose over 12 days. The BrdU labeling was analyzed by flow cytometry using the BrdU-FITC labeling kit (BD Biosciences). For in vitro BrdU incorporation, 3 μg/ml of BrdU was added to the culture and after five days the percentage of incorporation was determined as above. Apoptosis was measured using the Annexin V detection kit (BD Biosciences) with DAPI staining. For in vitro proliferation assays, SSCs were detached with 0.5% trypsin and loaded at $3 \times 10^4$ cells/well with cell trace violet (CTV, Thermofisher) for 15 min at 37 °C. CTV dilution was assessed by flow cytometry. To estimate the doubling time values, SSCs were seeded at $3 \times 10^3$ cells/cm² and counted after 3 days of culture. The doubling time was calculated as follows: (time of culture × log(2))/(log(final number of SSC) − log(initial number of SSC)).

## Quantitative real-time PCR

For mouse gene expression, total RNA was isolated from cultured or sorted SSCs and OPCs using the RNeasy Plus Mini or Micro Kit (Qiagen) and reverse transcribed with oligo(dT) and SuperScript II Reverse Transcriptase (Invitrogen). Quantitative RT-PCR reactions were performed on a Light Cycler instrument (LC480, Roche Diagnostics) with the LightCycler 480 SYBR Green detection kit (Roche Diagnostics) using primers reported in Table S3. For human gene expression, total RNA was isolated from cultured BMSCs using Trizol Reagent

(ThermoFisher). Reverse transcription was performed using SuperScriptVilo IV (ThermoFisher). When required, total RNA from WS BMSCs and their related controls were extracted from $0.2 \times 10^3$ BMSCs and pre-amplified using CellsDirect One-Step qRT–PCR kit (Invitrogen). PCR reactions were performed using primers reported in Table S3 with Power SYBRGreen (Applied Biosystems) on a 7500 FAST apparatus (Applied Biosystems). Mouse β-actin and 36b4 and human β-ACTIN and GAPDH were used as standards for normalization. Relative quantification was determined by the comparative delta-Ct ($2^{-\Delta CT}$) method (fold changes calculated relative to house-keeping genes) or delta-delta-Ct ($2^{-\Delta\Delta CT}$) method (fold changes calculated by setting the mean values obtained from WT cells as one).

## Multiplex qPCR

Multiplex qPCR was performed using the microfluidic Biomark system. One hundred SSCs were sorted into PCR tubes containing 5 μl of reverse transcription/pre-amplification mix containing 2X reaction buffer, SuperScriptIII from the CellsDirect One-Step qRT–PCR kit, and 0.2X Taqman assay (Life Technologies) (Table S4). cDNA pre-amplification was performed during 22 cycles and pre-amplified product was diluted 1:5 in TE buffer before processing with Dynamic Array protocol (Fluidigm). Cells expressing β-actin and control genes (Runx2, Col1α, Alp, and Ibsp) and not Pax5 and/or Cd3 (negative controls) were considered for analyses. Expression of β-actin was used for normalization. Heatmaps were generated with http://www.heatmapper.ca using Z scores.

## RNA sequencing

Pools of $3 \times 10^3$ SSCs or OPCs were sorted from the bone fraction into RLT buffer (Qiagen) with 1% of β-mercaptoethanol. RNA was isolated using RNeasy Micro Kit. cDNAs were generated from 400 to 1000 pg of total RNA using Clontech SMART-Seq v4 Ultra Low Input RNA kit for Sequencing (Takara Bio Europe) and amplified with 12 cycles of PCR by Seq-Amp polymerase. For Tn5 transposon tagmentation, 600 pg of pre-amplified cDNAs were used by the Nextera XT DNA Library Preparation Kit (96 samples) (Illumina) followed by library amplification of 12 cycles. Purification was performed with Agencourt AMPure XP and SPRIselect beads (Beckman-Coulter). Sequencing reads were generated, in Paired-End mode, on the GenomEast platform (Illumina). FastQC program was used to evaluate the quality of the raw sequencing data and reads shorter than 50 bp were removed. Reads were aligned to the Mus musculus genome (mm10 build) using the Star tool[90]. Gene expression quantification was obtained using read counting software Htseq[91]. Normalization and differential analysis were carried out with DESeq2 package by applying the Benjamini–Hochberg FDR correction ($p < 0.05$; 1.5-fold) for comparison between samples. Heatmaps and volcano plots were obtained using the web server Heatmapper and EnhancedVolcano packages, respectively.

## Statistics

All statistical analyses were conducted using Prism software (GraphPad Software). A Kruskal–Wallis test was used to determine the significance of the difference between means of WT, +/1013, and 1013/1013 groups. Unless specified, the unpaired two-tailed Student t test was used to compare means among two groups.

## Reporting summary

Further information on research design is available in the Nature Portfolio Reporting Summary linked to this article.

## Data availability

RNA-seq data that support the findings of this study have been deposited in the Gene Expression Omnibus repository with the accession code GSE217422. The data that support the findings of this

study are available from the corresponding author upon request. Source data are provided with this paper.

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

## Acknowledgements

We thank ML. Aknin (IPSIT, Facility PLAIMMO, Orsay), F. Mercier-Nomé (IPSIT, Facility PHIC, Orsay), Drs. V. Parietti-Montcuquet, C. Doliger, S. Duchez and N. Setterblad (Animal and Flow Cytometry Core Facilities, Institut de Recherche Saint-Louis, Paris), V. Nicolas (IPSIT, Facility MIPSIT, Orsay), D. Courilleau (IPSIT, Facility CIBLOT, Orsay) and C. Cordier and J. Megret (Plateau technique de cytométrie, SFR Necker, Paris) for their technical assistance. We thank the Montpellier Preclinical Platform of the Research Infrastructure ECELLFRANCE for the microCT analyses as well as the Plateforme d'Irradiation (IRSN, Fontenay-Aux-Roses, France) for their technical assistance. The study was supported by the LabEx LERMIT supported by ANR grant ANR-10-LABX-33 under the Program "Investissements d'Avenir" ANR-11-IDEX-0003-01, an ANR PRC grant (ANR-17-CE14-0019) to M.A.-L., C.B.-W. and coordinated by K.B., the FRM (Programme Equipe FRM 2022, EQU202203014627) and by the Association Saint Louis pour la Recherche sur les Leucémies to K.B. J.N. was a PhD fellow from the DIM Cancéropôle and the FRM. Z.A.N. was a fellowship recipient from the French Ministry and from the FRM (FDT202204015088). V.R. was supported by the FRM, La Ligue Contre le Cancer and la Société Française d'Hématologie. A.Bon. was supported by an ANR @RAction grant (ANR-14-ACHN-0008) and by a JCJC ANR grant (ANR-19-CE15-0019-01) to M.E. M.K. was a fellowship recipient from the French Ministry. A.Bou. was supported by the ANR grant 17-CE14-0019. J.P.L. was recipient from the People Program (Marie Curie Actions) of the European Union's Seventh Framework Program (FP7/2007-2013) under REA grant agreement n. PCOFUND-GA-2013-609102, through the PRESTIGE Program coordinated by Campus France, and from an ANR grant (ANR-17-CE14-0019). V.B., N.D., and K.B. were supported by the INCa agency under the program PRT-K 2017. J.K. was supported by European Union's Horizon 2020 MSCA, Program under grant agreement 641833 (ONCORNET). D.H.M. and P.M.M. were supported by the Division of Intramural Research of the National Institute of Allergy and Infectious Diseases, National Institutes of Health. Graphical abstract and mouse icons were created using the Biorender software (Biorender.com).

## Author contributions

A.A., J.N., and Z.A.N. designed and performed experiments, analyzed data, and contributed to manuscript writing; V.R., A.Bon., M.K., A.Bou., J.P.L., V.B., J.K., L.H.D.S., A.P., and A.C. performed experiments and analyzed data; S.P., N.D., M.A.-L., S.J.C.M., G.L., F.G., C.B.-W., and M.C.-S. performed experiments, contributed to data analyses, and reviewed the manuscript; D.H.M. and P.M.M. provided WS samples and clinical data and reviewed the manuscript; M.E. and M.R. helped with the study design, performed experiments, contributed to data analyses, and reviewed the manuscript; K.B. conceived, designed and supervised the study, contributed to data analyses, found funding for the study, and wrote the manuscript.

## Competing interests

The authors declare no competing interests.
