## [Peer Review File · Nature Communications]

WHIM Syndrome-linked CXCR4 mutations drive osteoporosisREVIEWER COMMENTS

Reviewer #1 (Remarks to the Author):

Anginot et al. report on the importance of CXCR4 desensitization in skeletal stem cells (SSCs) in order to allow SSCs to proliferate adequately and differentiate into the osteogenic lineage, whereas chondrogenic and adipogenic differentiation seem not to be affected by gain of function mutation of CXCR4. These novel data certainly increase our understanding on CXCR4 signalling in osteogenic lineage cells. In addition, the authors combined numerous well-designed in vivo and in vitro experiments to elucidate the cellular mechanisms. However, several inconsistencies between the findings are present, especially concerning the effect of CXCR4 desensitization on SSC properties and their osteogenic differentiation potential.

Figure 2B. The decrease in stroma cell number in mutant mice (60×10^3 versus 100×10^3 in WT mice) cannot be explained by the combined decrease in SSC (3×10^3 versus 5×10^3) and OPC (7×10^3 versus 13×10^3). The question is therefore which other bone cell types are decreased in mutant mice as these other cell types might also contribute to the observed decrease in bone mass. Are endothelial cells decreased (H-type and L-type) in mutant mice as they express CXCR4 and might provide a vascular niche for the SSC?

Figure 2N and 2S. Parameters of TBV and cortical bone should be quantified, preferable by μ CT (or quantitative histological analysis). At this moment, only 1 image per condition is shown and this is an Opn staining, which is not considered to be the appropriate approach for quantitative bone measurements. This quantification of bone parameters is especially necessary to verify the bone loss that occurs when recipient mice are WT (Figure 2S), as the bone loss that is induced by transplantation of mutant donor cells in WT recipient mice is hard to be explained only by a reduced number of SSC and OPC, as is now suggested.

Figure 3. The authors suggest that the in vivo observed increase in osteoclasts in mutant mice is linked to an altered BM environment. To strengthen this statement, coculture experiments of osteogenic cells and osteoclast-precursors, in different combinations of WT vs mutant cells (treated with PTH, Pg or 1,25-vitamin D) should be performed. At this moment, the data only describe a discordance between the in vivo and in vitro findings, but do not allow to make any conclusion on whether the decrease in bone mass is partly caused by increased bone resorption.

Figure 3. The histomorphometric data should be confirmed in more mice, as 3 mice per group for histomorphometric analysis is often not sufficient (Figure 3F-H). In addition, the bone formation data are puzzling, as osteoblast surface and osteoid surface are normal, but MS/BS and DB/BS are decreased. The authors interpret these data as a 'lower number of osteoblasts' but this statement does not fit with the normal osteoblast surface that is observed. Since dynamic bone formation parameters primarily measure the incorporation of minerals, these data might suggest that the formation of bone matrix by the osteoblasts is normal, but that the mineralisation of this bone matrix is impaired (and likely some osteoblasts are not mineralizing the matrix, whereas others show normal mineralization capacity as MAR is normal). Gene expression analysis might provide some more insight. The gene expression analysis (Figure 3J) is now restricted to genes that typify mature osteoblasts, but the expression of genes involved in mineralization is not analysed. In addition, the

variation of the gene expression data reported in Figure 3J and M, is rather high and this quantification should be validated with qRT-PCR data and using more mice.

Figure 4. Panel D shows the relative expression of selected genes; are these the most differentially expressed genes between the different genotypes? To appreciate the importance of these differences, it will be important to provide also the unbiased ranked overview/list of pathways which differ the most between genotypes, based on genes involved. In addition, since mutant SSC maintain their potential to differentiate normally to chondrocytes and adipocytes, are SOX9 and Pparg expression normal in mutant SSC? Furthermore, it remains hard to understand that a decrease in OPC number (Figure 2B, Figure 4I) does not affect osteoblast or osteoid surface. How do the authors reconcile these data?

Figure 5 and 3. The data suggest that osteogenic differentiation starting from mutant SSC is reduced (Figure 5E-G), but once mutant SSC become OPC they can differentiate normally (Figure 3L). It should be good to confirm this observation, by performing the same assays on OPC as shown for SSC (Figure 5: differentiation with Alp quantification and gene expression analysis). In addition, it is rather particular that after 21 days of osteogenic differentiation, most of the cells are still SSC (Figure S1), and intermediate cells account only for 15% of the population, whereas the % of ALP+ cells, reported in Figure 5E, seems much higher. Same comment for the low % of mature cells compared to reported homogeneous and abundant alizarin red staining (Figure 5F).

Figure 5 in vivo data. The authors state that especially the cortical bone is rescued in mutant mice, but not the trabecular phenotype, based on lumbar spine BMD data. To validate this statement, μ CT analysis of cortex of long bones should be analysed with and without AMD3100 treatment. In addition, these data also suggest that CXCR4 desensitisation in osteogenic lineage cells is likely not responsible for the trabecular bone phenotype, and that other cell types/mechanisms might be involved. This site-specificity should be reflected in the title and in the abstract.

Minor comments

Perilipin staining should be quantified as the observation that CXCR4 specifically reduces the osteogenic, but not the adipogenic differentiation is interesting, but should be validated by quantitative data.

Figure 1D: it is not clear whether the total number of mice used is 7-14, coming from 3 experiments, or that in each of the 3 experiments there were 7-14 mice, thus 21-42 mice in total. Similar comment to all experiments using mice.

Figure 2J: the % of apoptotic OPC is around 30%, which is rather high, and should be commented on.

Reviewer #2 (Remarks to the Author):

WHIM syndrome (WS) is a rare immunodeficiency caused by gain-of-function CXCR4 mutations. The authors have demonstrated for the first time a substantial decrease in bone mineral density in 25% of WS patients and osteoporosis in a WS mouse model. Interestingly, wild-type mice transplanted with bone marrow hematopoietic cells from mice with a WS-linked CXCR4 mutation (*Cxcr4*^{+/1013} or *Cxcr4* 1013/1013) had reduced trabecular bone content compared with wild-type chimeras. On the other hand, transplantation of wild-type bone marrow cells did not rescue the reduced trabecular bone content in the mutant chimeras. Osteogenic differentiation of cultured bone marrow skeletal stem cells (SSCs) from the mutants was impaired *in vitro*. The CXCR4 antagonist AMD3100 normalized *in vitro* osteogenic potential of SSCs and reversed an *in vivo* decrease in Sca-1-PDGFRa⁻ cells in the mutants. These results are interesting and important; however the major concern remains at this time. There is the possibility that osteopenia in mice and patients, which carry the WS-linked CXCR4 mutation, is the result of only enhanced osteoclast function but not reduced osteogenic differentiation of SSCs.

1. As the authors described, it has been shown previously that deletion of CXCR4 in mesenchymal cells, including SSCs, resulted in osteopenia (Tzeng et al., *J. Bone Miner. Res.* 2018; Zhu et al., *J. Biol/Chem.* 2011). These results argue against the authors' conclusion that gain-of-function CXCR4 mutations in SSCs resulted in osteopenia. Thus, I would recommend the authors to generate and analyze the mice, in which mesenchymal cells, including SSCs, but not hematopoietic cells carry WS-linked CXCR4 mutations.

2. The authors show the reduced trabecular bone content of mice with a WS-linked CXCR4 mutation transplanted with bone marrow hematopoietic cells from wild-type mice was not rescued 3 and 16 weeks after transplantation. However, wild-type hematopoietic cells might be able to rescue the reduced trabecular bone content of the mutants earlier in development.

3. The authors describe Sca-1⁺PDGFRa⁺ cells as SSCs (Page 8, line 142); however, the major population of bone marrow SSCs is defined as Sca-1⁻PDGFRa⁺PDGFRb⁺LepR⁺CD31⁻ cells (Omatsu et al., *Immunity* 2010; Zhou et al., *Cell Stem Cell* 2014; Seike et al., *Genes Dev* 2018).

4. The evidence that Sca-1⁻PDGFRa⁻ cells are committed osteoblasts (OPCs) in the bone marrow would not be convincing (Page 8, line 143).

Reviewer #4 (Remarks to the Author):

The manuscript by Anginot and colleagues provides novel insights into the role of CXCR4-mediated signaling in skeletal stromal/stem cell osteogenic specification. The authors describe a series of experiments characterizing the anatomic, developmental and functional properties of the skeletal and osteogenic compartment in a knock-in mouse model of the human genetic disorder WHIM syndrome. The significance of the deficits in skeletal remodeling and stem cell differentiation identified in the mouse model in human bone biology are validated in cohort of WHIM syndrome patients carrying gain-of-function mutations in CXCR4. These findings represent a novel contribution elucidating an important new role for CXCR4 in bone biology.

The authors characterized the effects of increased CXCR4 signaling in vivo through standard histomorphometric of bone anatomy and flow cytometric analyses of various progenitor cell populations in the mouse model. The data in Figure 1 are well presented and convincing in regard to the gene-dose dependent skeletal effects as well as the specificity of the changes to cortical and trabecular bone. Figure 2 is overly dense and contains information that could be moved to the supplement without impacting the major findings of the work. In particular, the experiments demonstrating the functional effects of the mutant CXCR4 receptor recapitulate characteristics of CXCR4 C-terminal truncations that have been well studied in other contexts. It would suffice to state that the mutant receptor localization, internalization and intracellular signaling were similar to what has been seen in other contexts and move panels 2E-J to the supplement. The bone marrow reconstitution experiments shown in the remainder of the figure demonstrate clearly the contribution of cell-extrinsic as well as cell-intrinsic factors to the observed skeletal changes. Similarly, the effects on bone resorption and formation shown in Figure 3 panels C-E can be moved to supplement to better focus on the transcriptional effects shown in the subsequent panels.

The data in Figures 4 and 5 provide compelling data regarding the impact of aberrant CXCR4 signaling on osteogenic specification at the level of transcriptional effects and cell cycle progression. The PCA data shown in Figure 4C is not well explained as the 48 genes used for expression profiling are not described in the text nor the supplement, which lists a smaller number of genes. The data in the subsequent panels are more informative. I would consider removing panel 4C or moving it to the supplement with a better description of the analysis. The experiments shown in Figure 5 document the selective reduction in osteogenic differentiation capacity of stromal stem cells carrying one or two mutant CXCR4 receptors in a dose dependent fashion and the reversal of this phenotype with treatment of the receptor inhibitor AMD3100. The relevance of these data in mice to human bone biology are supported with the data shown in Figure 6 which revealed a selective osteogenic differentiation defect in bone marrow cells derived from WHIM syndrome patients.

With regards to the conclusion that a skeletal phenotype is present in a subset of WHIM syndrome patients, given that treatment of neutropenia with G-CSF is associated with osteopenia as side effect of therapy, it would be useful to know the total number of patients treated with G-CSF in the cohort to address the concern that the enrichment in osteopenic patients is restricted to those patients that have been so treated as well as their ages and genders given the impact of these variables on risk for osteopenia in general.

Apart from these concerns, the quality of the data presented is good and the conclusions supported by the evidence. The manuscript is well written and the references appropriate, though it was notable that the initial description of the cause of WHIM syndrome as gain-of-function truncation mutations in CXCR4 was not cited, this should be added.

Point-to-point response to the reviewers' comments

Reviewer #1 comments:

Anginot et al. report on the importance of CXCR4 desensitization in skeletal stem cells (SSCs) in order to allow SSCs to proliferate adequately and differentiate into the osteogenic lineage, whereas chondrogenic and adipogenic differentiation seem not to be affected by gain of function mutation of CXCR4. These novel data certainly increase our understanding on CXCR4 signalling in osteogenic lineage cells. In addition, the authors combined numerous well-designed in vivo and in vitro experiments to elucidate the cellular mechanisms. However, several inconsistencies between the findings are present, especially concerning the effect of CXCR4 desensitization on SSC properties and their osteogenic differentiation potential.

Major concerns:

1. **“Figure 2B. The decrease in stroma cell number in mutant mice (60×10^3 versus 100×10^3 in WT mice) cannot be explained by the combined decrease in SSC (3×10^3 versus 5×10^3) and OPC (7×10^3 versus 13×10^3). The question is therefore which other bone cell types are decreased in mutant mice as these other cell types might also contribute to the observed decrease in bone mass. Are endothelial cells decreased (H-type and L-type) in mutant mice as they express CXCR4 and might provide a vascular niche for the SSC?”**

We are grateful to the reviewer for this constructive comment and agree that some populations are likely missing in our flow-cytometric analyses. In particular, we did not consider the CD51-Sca1- cell population which is non-hematopoietic (CD45-) and non-vascular (CD31-) but appeared to be decreased in an allele-dose dependent manner in mutant mice. Thus, this stromal population might contribute to the overall decrease in stroma cell number in mutant mice. Because we do not know anything about this population, we propose to remove the stroma quantification panel to better focus on SSCs and OPCs (new Figures 2A and 2B). Whether endothelial cell (EC) numbers are affected is an interesting question raised by the reviewer. Different types of bone marrow (BM) ECs have been phenotypically identified in long bones (see for instance Kusumbe Nature 2014; Balzano Cell Rep 2019). The bone fraction is reported to be enriched for arteriolar ECs (Sca1+CD31+Emcn-), few L-type sinusoidal ECs and CD31hiEmcnhi H-type ECs, a small fraction of the ECs at the end of the CD31+Emcn- arteriolar network. Based on Sca1 and CD31 expression, we observed by flow cytometry a decrease in ECs in the bone fraction of mutant mice (see below Figure 1 for reviewers). Although these preliminary findings are very interesting, we feel that they deserve to be strengthened by adding notably the Endomucin marker to visualize by immunofluorescence the impact of the *Cxcr4* mutation on H-type and L-type EC architecture and numbers. This would constitute the subject of another study that will be dedicated to vascular modifications in WS mice. However, the well-established regulatory role of the vascular system on the mesenchymal one has been discussed in the revised version of the manuscript (page 22, lines 474 and 482). In particular, whether vascular cells participate in the defective osteolineage specification of SSCs in *Cxcr4*¹⁰¹³-bearing mice deserves further investigations.

Figure 1: Reduced endothelial cells in the bone fraction of mutant mice. Absolute numbers of the indicated stroma cell subsets from bone fractions were determined by flow cytometry in WT, +/1013 and 1013/1013 mice. Data (means + SEM) are from three independent experiments with 6 mice in total per group. *, P < 0.05; and **, P < 0.005 compared with WT cells. §§, P < 0.005 compared with +/1013 cells. (as determined using the two-tailed Student's t test).

2. “Figure 2N and 2S. Parameters of TBV and cortical bone should be quantified, preferable by μ CT (or quantitative histological analysis). At this moment, only 1 image per condition is shown and this is an Opn staining, which is not considered to be the appropriate approach for quantitative bone measurements. This quantification of bone parameters is especially necessary to verify the bone loss that occurs when recipient mice are WT (Figure 2S), as the bone loss that is induced by transplantation of mutant donor cells in WT recipient mice is hard to be explained only by a reduced number of SSC and OPC, as is now suggested.”

We are grateful to the reviewer for this helpful suggestion and as requested, we have quantified trabecular and cortical bone parameters by μ CT (new Figures 20 and 2P and new supplemental Figure 1G). By this way, we confirmed the bone loss in WT recipient upon transplantation of mutant BM, thereby indicating cell-extrinsic (hematopoietic) *Cxcr4*-mediated regulation of the skeletal landscape. The text has been modified accordingly (page 11, line 230; page 9, line 183). One can speculate that myeloid cells including OCLs as well as lymphoid cells may actively participate in promoting bone remodeling in BM chimeric WT recipient mice. Indeed, the laboratory of Pr. A. Bozec among others recently reported that prolonged HIF-1 α signaling in B cells leads to enhanced RANKL production and OCL formation in the BM (Meng *et al.*, Bone Research 2022). Likewise, BM T cells are known to produce RANKL and to regulate OCL compartment within the BM (see for review for instance Corrado *et al.*, IJMS 2020; Mori *et al.*, Clin Dev Immunol 2013; Zhang *et al.*, Front Endocrinol 2020). Whether the transplantation of mutant BM recreates a pro-osteoclastogenic environment through a remodeling of the lymphoid compartment deserves further investigations. This point has now been discussed in the revised version of the manuscript (page 21, line 459).

3. “Figure 3. The authors suggest that the *in vivo* observed increase in osteoclasts in mutant mice is linked to an altered BM environment. To strengthen this statement, coculture experiments of osteogenic cells and osteoclast-precursors, in different combinations of WT vs mutant cells (treated with PTH, Pg or 1,25-vitamin D) should be performed. At this moment, the data only describe a discordance between the *in vivo* and *in vitro* findings, but do not allow to make any conclusion on whether the decrease in bone mass is partly caused by increased bone resorption.”

We sincerely thank the reviewer for this very relevant and helpful comment. We fully agree with the point that making a link between osteogenic cells and osteoclast precursors is of importance. As recommended by the reviewer, we addressed it using a co-culture system between *in vitro* expanded osteogenic cells carrying or not the *Cxcr4* mutation and WT OCL precursors, *ie.*, BM CD11b⁺ myeloid cells. As shown in the new Figure 3L, mutant osteogenic cells promoted exacerbated OCL differentiation compared to WT cells. Soluble factors seem to be not sufficient to explain this bias as the supernatants of stimulated expanded osteogenic cells (WT or mutant) did not induce OCL differentiation. Additionally, transcriptomic analyses of stimulated osteogenic cells carrying or not the *Cxcr4* mutation did not reveal any major changes in expression levels of master genes regulating osteoclastogenesis such as the RANKL/OPG balance or the M-Csf cytokine (see new Figure 3M). These findings suggest a juxtacrine function of osteogenic cells toward OCL differentiation that likely relies on direct interactions between both cell types and involves the *Cxcl12/Cxcr4* axis. As adding the osteogenic component carrying the *Cxcr4* mutation is sufficient to promote *in vitro* enhancement of OCL differentiation, we propose that the overall decrease in bone mass in mutant mice involves remodeling of osteogenic and osteoclastogenic components leading to decreased bone formation and increased bone resorption. Although the use of a conditional mutant mouse model would be the ideal way to confirm these findings, such a model is not currently available to our knowledge. In such a process, the osteogenic lineage would act as the driver and the OCL one as a passenger. The underlying molecular mechanism(s)

of this cross-talk remains to be elucidated, but seems to require direct contact between both cell types. The text has been modified accordingly (page 11, line 230; page 21, line 451).

4. “Figure 3. The histomorphometric data should be confirmed in more mice, as 3 mice per group for histomorphometric analysis is often not sufficient (Figure 3F-H). In addition, the bone formation data are puzzling, as osteoblast surface and osteoid surface are normal, but MS/BS and DB/BS are decreased. The authors interpret these data as a ‘lower number of osteoblasts’ but this statement does not fit with the normal osteoblast surface that is observed. Since dynamic bone formation parameters primarily measure the incorporation of minerals, these data might suggest that the formation of bone matrix by the osteoblasts is normal, but that the mineralisation of this bone matrix is impaired (and likely some osteoblasts are not mineralizing the matrix, whereas others show normal mineralization capacity as MAR is normal). Gene expression analysis might provide some more insight. The gene expression analysis (Figure 3J) is now restricted to genes that typify mature osteoblasts, but the expression of genes involved in mineralization is not analysed. In addition, the variation of the gene expression data reported in Figure 3J and M, is rather high and this quantification should be validated with qRT-PCR data and using more mice.”

As requested by the reviewer, histomorphometric and osteoclast data have been implemented by adding two to three mice per group. These results that are now displayed in Figure 3A-3E confirmed the previous ones, *ie.* increased OCL surface and number and decreased total and double labelled bone surfaces in mutant mice compared to WT ones. Mineral apposition rate was similar in WT and *Cxcr4*¹⁰¹³-bearing mice, while bone formation rate is lower in mutant mice. These data prompt us to suggest a decrease in bone formation related to a lower number of OBLs with maintained activity of each individual OBL. In line with preserved intrinsic bone formation capacities of active osteoblastic lineage cells in mutant mice, our RNA-seq analyses of bulks sorted from the bone fraction highlighted in mutant OPCs a gene signature with preserved mineralized matrix potential that has been confirmed by qPCR analyses (see new Figures 3F-H and S1K-M). In agreement, sorted OPCs from mutant mice were as efficient as WT ones *in vitro* at producing differentiated OBLs and mineralized nodules after 14- or 21-days culture in osteogenic medium as determined by Alkaline phosphatase and Alizarin Red staining respectively (see new Figures 3I and S1N). This was confirmed by qPCR analyses with no changes in expression of genes encoding osteogenic regulators in mutant cultures (see new Figure S1O). These findings are in line with efficient terminal osteogenic differentiation and preserved bone formation and mineralization capacities in *Cxcr4*¹⁰¹³-bearing mice. The text has been modified accordingly (page 11, lines 208 & 219).

5. “Figure 4. Panel D shows the relative expression of selected genes; are these the most differentially expressed genes between the different genotypes? To appreciate the importance of these differences, it will be important to provide also the unbiased ranked overview/list of pathways which differ the most between genotypes, based on genes involved. In addition, since mutant SSC maintain their potential to differentiate normally to chondrocytes and adipocytes, are SOX9 and Pparg expression normal in mutant SSC? Furthermore, it remains hard to understand that a decrease in OPC number (Figure 2B, Figure 4I) does not affect osteoblast or osteoid surface. How do the authors reconcile these data?”

We sincerely thank the reviewer for bringing to our attention that unbiased transcriptomic analyses of WT and mutant SSC are needed. As requested, we investigated the impact of the gain-of-*Cxcr4*-function on the molecular identity of SSCs by performing RNA-seq analyses of sorted bulk cells from WT and mutant bone fractions. Biological processes related to cell cycle and osteogenic differentiation were significantly modulated in 1013/1013 SSCs compared to WT SSCs as determined by GSEA (Gene set enrichment analysis) (see new Figure 4C). The gene signature related to cell cycle progression and regulation was reduced in 1013/1013 SSCs compared to WT

ones (see new Figures S2A and S2B). Likewise, genes related to osteogenic differentiation appeared to be decreased in mutant SSCs (see new Figures 4D and 4E). In contrast, key genes involved in both adipogenesis and chondrogenesis were not differentially expressed in mutant SSCs (see new Figure S2C). These results were confirmed by microfluidic-based multiplex gene expression analyses (see Figures 4F and 4G and Figure S2D), thus suggesting that proper Cxcr4 signaling is required for regulating osteogenic specification of SSCs at the transcriptional level. The text has been modified accordingly (page 13, line 264). Regarding the last point about our flow-cytometric and histomorphometric results, we agree that decreased OPC number cannot fully be explained by the unremarkable osteoid and osteoblast number. We therefore measured the labelled surfaces and MAR and also calculated the bone formation rate, which are more accurate indices of dynamic bone formation. Indeed, labelled surfaces and bone formation rate are decreased, which is in favor of reduced OBL differentiation, while the MAR remained identical, thus suggesting a maintained capacity of osteoblast to produce matrix once differentiated.

6. “Figure 5 and 3. The data suggest that osteogenic differentiation starting from mutant SSC is reduced (Figure 5E-G), but once mutant SSC become OPC they can differentiate normally (Figure 3L). It should be good to confirm this observation, by performing the same assays on OPC as shown for SSC (Figure 5: differentiation with Alp quantification and gene expression analysis). In addition, it is rather particular that after 21 days of osteogenic differentiation, most of the cells are still SSC (Figure S1), and intermediate cells account only for 15% of the population, whereas the % of ALP+ cells, reported in Figure 5E, seems much higher. Same comment for the low % of mature cells compared to reported homogeneous and abundant alizarin red staining (Figure 5F).”

We are grateful to the reviewer for this constructive suggestion and as requested we performed Alp quantification and gene expression analyses as already explained in response to the point#4 above. Our novel data showed that sorted OPCs from mutant mice were as efficient as WT ones *in vitro* at generating bone-making OBLs after 14-days culture in osteogenic medium as determined by Alp staining (see new Figure S1N). This was further confirmed by qPCR analyses with no changes in expression of genes encoding osteogenic regulators in mutant cultures (see new Figure S1O). These findings are in line with efficient terminal osteogenic differentiation and preserved bone formation and mineralization capacities in *Cxcr4*¹⁰¹³-bearing mice. The text has been modified accordingly (page 11, line 219). We fully agree with the reviewer that the yield of immature and mature osteogenic cells recovered by flow cytometry was not as high as expected in light of Alp and Alizarin red staining, and this was likely due to the difficulty we experimented to collect and separate homogeneously the cells from the mineralized matrix at the end of the culture. Although real-time quantitative PCR analyses of Sca-1 and PDGFR α markers corroborated the flow cytometric results (see Figure S2E), these flow cytometric results are rather dispensable for the paper and therefore we propose to remove them to clarify the message. We thank the reviewer for having pointed this inconsistency.

7. “Figure 5 in vivo data. The authors state that especially the cortical bone is rescued in mutant mice, but not the trabecular phenotype, based on lumbar spine BMD data. To validate this statement, μ CT analysis of cortex of long bones should be analysed with and without AMD3100 treatment. In addition, these data also suggest that CXCR4 desensitisation in osteogenic lineage cells is likely not responsible for the trabecular bone phenotype, and that other cell types/mechanisms might be involved. This site-specificity should be reflected in the title and in the abstract.”

We thank the reviewer for this relevant comment. Our original version of the manuscript stated a suggestion for a correcting effect of Cxcr4-dependent signaling dampening on the cortical, rather than trabecular, bone based on BMD values of lumbar spine in mutant mice. Because μ CT analyses were not carried out for this experiment, we sought to measure the cortical thickness in paraffin-embedded sections stained with Toluidine Blue. The two cortices were measured using

histomorphometry software and expressed as mean of both cortices for each sample. As shown in the new Figure 5L, AMD3100 treatment for 3 weeks did not ameliorate the cortical network in mutant mice, thus suggesting that either the treatment procedure should be further optimized in terms of duration and kinetics or, as anticipated by the reviewer, that other cell types/mechanisms might be involved at this stage such as OCLs or perivascular SSCs as recently reported by Jeffery and coll. (Cell Stem Cell, 2022). This warrants further investigations. This point has been mentioned in the revised version of the manuscript (page 16, line 345).

Minor concerns:

1. “Perilipin staining should be quantified as the observation that CXCR4 specifically reduces the osteogenic, but not the adipogenic differentiation is interesting, but should be validated by quantitative data.”

We thank the reviewer for bringing to our attention that quantification data would be helpful to strengthen the significance of our findings. As requested, Figure 1H mentioned by the reviewer has been edited with quantification data and shows no change in adipocyte content in the BM of mutant mice, as compared to WT mice (see new Figure 1J). Congruent with immunostaining on bone sections, RNA-seq analyses performed during the reviewing period show that mutant SSCs displayed a gene signature consistent with preserved adipogenic potential (see new Figures S2C and S2D). These cells also differentiated into adipocytes similarly to WT SSCs when cultured *in vitro* in adipogenic medium (Figure S2G). These results suggest that proper Cxcr4 signaling is required for regulating the osteogenic specification of SSCs specifically. The text has been modified accordingly (page 7, line 130; page 13, line 264, page 15 line 316).

2. “Figure 1D: it is not clear whether the total number of mice used is 7-14, coming from 3 experiments, or that in each of the 3 experiments there were 7-14 mice, thus 21-42 mice in total. Similar comment to all experiments using mice.”

We thank the reviewer for bringing to our attention that the total number of mice used in each experiment was not clear and we apologize for that. In fact, each number mentioned represents the total number of mice used, in 3 independent experiments or more. The legends have been modified accordingly.

3. “Figure 2J: the % of apoptotic OPC is around 30%, which is rather high, and should be commented on.”

We thank the reviewer for this relevant comment. Although the reason why the apoptosis rate is high among OPCs is unclear, we obtained similar results using cleaved caspase 3 staining instead of Annexin V staining. One can speculate that experimental procedures make these cells more fragile and prone to undergo apoptosis. In both assays, we were unable to observe differences between WT and mutant OPCs, thus strongly suggesting that increased apoptosis of OPCs does not contribute to bone loss in mutant mice. As requested by the Reviewer#4, this panel has been moved to the supplemental Figure 1 (see Figure S1F).

Reviewer #2 comments:

WHIM syndrome (WS) is a rare immunodeficiency caused by gain-of-function CXCR4 mutations. The authors have demonstrated for the first time a substantial decrease in bone mineral density in 25% of WS patients and osteoporosis in a WS mouse model. Interestingly, wild-type mice transplanted with bone marrow hematopoietic cells from mice with a WS-linked CXCR4 mutation (Cxcr4+/1013 or Cxcr4 1013/1013) had reduced trabecular bone content compared with wild-type chimeras. On the other hand,

transplantation of wild-type bone marrow cells did not rescue the reduced trabecular bone content in the mutant chimeras. Osteogenic differentiation of cultured bone marrow skeletal stem cells (SSCs) from the mutants was impaired in vitro. The CXCR4 antagonist AMD3100 normalized in vitro osteogenic potential of SSCs and reversed an in vivo decrease in Sca-1-PDGFRa- cells in the mutants. These results are interesting and important; however the major concern remains at this time. There is the possibility that osteopenia in mice and patients, which carry the WS-linked CXCR4 mutation, is the result of only enhanced osteoclast function but not reduced osteogenic differentiation of SSCs.

Major concerns:

1. “As the authors described, it has been shown previously that deletion of CXCR4 in mesenchymal cells, including SSCs, resulted in osteopenia (Tzeng et al., J. Bone Miner. Res. 2018; Zhu et al., J. Biol/ Chem. 2011). These results argue against the authors’ conclusion that gain-of-function CXCR4 mutations in SSCs resulted in osteopenia. Thus, I would recommend the authors to generate and analyze the mice, in which mesenchymal cells, including SSCs, but not hematopoietic cells carry WS-linked CXCR4 mutations.”

We are grateful to the reviewer for this relevant and constructive comment. Indeed, truncating mutations in CXCR4 which cause the WHIM syndrome (WS) in humans lead *in vitro* to a typical gain-of-function response to CXCL12 as exemplified by enhanced chemotaxis. However, in several cellular contexts (e.g., HSC lymphoid differentiation, B cell development...), we observed that loss of CXCR4 and gain of function of CXCR4 translated into similar phenotypes. This likely relates to the intensity and the strength of CXCR4 signaling that should be tightly regulated to permit the occurrence of physiological functions. Our findings unveil that mutant SSCs from the bone fraction are impaired in their capacities to generate OBLs as illustrated notably *in vitro* thus implying a cell-autonomous effect of the *Cxcr4* mutation in the bone phenotype. In line with this, J. Pereira’s laboratory recently showed using a second mouse model of the WS, carrying the gain-of-function CXCR4 R334X mutation, that lymphopoiesis is reduced because of a dysregulated transcriptome of mesenchymal stem cell isolated from the flushed marrow fraction and characterized by a switch from an adipogenic to an osteolineage-prone program with limited lymphopoietic activity (Zehentmeier et al., Science Immunology 2022). These results agree with ours and suggest that both hematopoietic and stromal cells are affected by the *Cxcr4* gain of function mutation. The text has been modified accordingly (page 5, line 90; page 19, line 403).

Our reciprocal BM reconstitution experiments support this assumption since transplantation of WT BM into lethally irradiated mutant recipients was not sufficient to rescue the skeletal landscape phenotype, and conversely, transplantation of mutant BM induced bone dysregulation in WT recipient (see Figures 3E-P and S1G). Although we are aware of the fact that BM chimera do not constitute perfect models, we do believe they are informative notably when hematopoietic cells that are engrafted do not carry WS-linked CXCR4 mutations. Moreover, we think that our ubiquitous mouse model is relevant since it closely phenocopies the immune-hematological phenotype of the human pathology in which both hematopoietic and stromal cells harbor the *Cxcr4* mutation. To confirm that, a conditional mouse model would have been ideal and not beyond the scope but we are not aware that such a model exists and it was not feasible *de novo* in the frame of a reviewing period. Rather, as suggested by the Reviewer#1, we set-up a co-culture system between *in vitro* expanded osteogenic cells carrying or not the *Cxcr4* mutation and WT OCL precursors, *ie.*, BM CD11b+ myeloid cells. As shown in the new Figure 3L, mutant osteogenic cells promoted exacerbated OCL differentiation compared to WT cells. Soluble factors do not seem sufficient as the supernatants of such stimulated expanded osteogenic cells (WT or mutant) did not induce OCL differentiation. Additionally, transcriptomic analyses of stimulated osteogenic cells carrying or not the *Cxcr4* mutation did not reveal any major changes in expression levels of master genes regulating osteoclastogenesis such as the RANKL/OPG

balance (see new Figure 3M). These findings suggest a juxtacrine function of osteogenic cells toward OCL differentiation that likely relies on direct interactions between both cell types and involves the Cxcl12/Cxcr4 axis. As adding the osteogenic component carrying the *Cxcr4* mutation is sufficient to promote *in vitro* enhancement of OCL differentiation, we propose that the overall decrease in bone mass in mutant mice involves remodeling of osteogenic and osteoclastogenic components leading to decreased bone formation and increased bone resorption. In such a process, the osteogenic lineage would act as the driver and the OCL one as a passenger. The underlying molecular mechanism(s) of this cross-talk remains to be elucidated but seems to require direct contact between both cell types. The entire manuscript as well as the title have been modified accordingly (page 11, line 230; page 21, lines 451 & 459) and a graphical abstract has been designed consequently.

2. “The authors show the reduced trabecular bone content of mice with a WS-linked CXCR4 mutation transplanted with bone marrow hematopoietic cells from wild-type mice was not rescued 3 and 16 weeks after transplantation. However, wild-type hematopoietic cells might be able to rescue the reduced trabecular bone content of the mutants earlier in development.”

We thank the reviewer for this relevant comment. However, we have to stress that currently we do not have the ethical authorization to transplant BM into mice younger than seven/eight weeks but we are aware that it would be interesting to do it. This has been clearly mentioned in the revised version of the manuscript (page 9, line 175).

3. “The authors describe Sca-1+PDGFRa+ cells as SSCs (Page 8, line 142); however, the major population of bone marrow SSCs is defined as Sca-1-PDGFRa+PDGFRb+LepR+CD31- cells (Omatsu et al., Immunity 2010; Zhou et al., Cell Stem Cell 2014; Seike et al., Genes Dev 2018).”

We are grateful to the referee for pointing out that the phenotype of SSCs we used could be a matter of debate and should be better justified. To the best of our knowledge, there is currently no consensual denomination for the different BM mesenchymal subpopulations and we agree with the reviewer that we should have been more precise on this point. As shown in the paper of Zhou *et al.* (Cell Stem Cell, 2014), the highest CFU-F clonogenic potential is observed in the Sca1+PDGFRa+ subset and not in the Sca1-PDGFRa+ population. This has been confirmed and extended to SSCs in the periosteum (Jeffery *et al.*, Cell Stem Cell, 2022). Furthermore, 16wks after transplantation of GFP+ Sca1+PDGFRa+ into WT mice (Morikawa *et al.*, JEM 2009), it was shown that among the GFP+ cells recovered, a few were Sca1+PDGFRa+ and most of them were Sca1-PDGFRa+, indicating that Sca1+PDGFRa+ cells are at the top of the hierarchy. This is why we chose to consider the Sca1+PDGFRa+ cells in the bone fraction as skeletal stem cells as compared to Sca1-PDGFRa+ that are more engaged in differentiation, as osteoblast progenitor cells. This point has been mentioned in the revised version of the manuscript (page 8, line 146).

4. “The evidence that Sca-1-PDGFRa- cells are committed osteoblasts (OPCs) in the bone marrow would not be convincing (Page 8, line 143).”

We apologize for the lack of clarity with this sentence. As explained in the point 3, we consider the CD51+Sca1- population as more differentiated than its Sca1+ counterpart and the sentence has been modified accordingly (page 8, line 146). There was also a typo and we should have referred to the CD51+Sca1- population as PDGFRa+/- as it includes both PDGFRa positive and negative subsets. In line with this, we already consider early OPCs with multipotent adipo/osteogenic potential in the flushed stromal marrow fraction as Sca-1-negative and PDGFRa-positive (see new Figure 4M). Our previous results showed that the Sca1-PDGFRa- population highly express committed osteoblast markers such as Bglap, Col1a1 and Pth1r1 (see Balzano *et al.*, Cell Reports 2019).

Reviewer #4 comments:

The manuscript by Anginot and colleagues provides novel insights into the role of CXCR4-mediated signaling in skeletal stromal/stem cell osteogenic specification. The authors describe a series of experiments characterizing the anatomic, developmental and functional properties of the skeletal and osteogenic compartment in a knock-in mouse model of the human genetic disorder WHIM syndrome. The significance of the deficits in skeletal remodeling and stem cell differentiation identified in the mouse model in human bone biology are validated in cohort of WHIM syndrome patients carrying gain-of-function mutations in CXCR4. These findings represent a novel contribution elucidating an important new role for CXCR4 in bone biology.

Major concerns:

1. “The authors characterized the effects of increased CXCR4 signaling in vivo through standard histomorphometric of bone anatomy and flow cytometric analyses of various progenitor cell populations in the mouse model. The data in Figure 1 are well presented and convincing in regard to the gene-dose dependent skeletal effects as well as the specificity of the changes to cortical and trabecular bone. Figure 2 is overly dense and contains information that could be moved to the supplement without impacting the major findings of the work. In particular, the experiments demonstrating the functional effects of the mutant CXCR4 receptor recapitulate characteristics of CXCR4 C-terminal truncations that have been well studied in other contexts. It would suffice to state that the mutant receptor localization, internalization and intracellular signaling were similar to what has been seen in other contexts and move panels 2E-J to the supplement. The bone marrow reconstitution experiments shown in the remainder of the figure demonstrate clearly the contribution of cell-extrinsic as well as cell-intrinsic factors to the observed skeletal changes. Similarly, the effects on bone resorption and formation shown in Figure 3 panels C-E can be moved to supplement to better focus on the transcriptional effects shown in the subsequent panels.”

We are grateful to the reviewer for these constructive suggestions and as requested, the panels 2E-J and 3C-E have been moved to the new Supplemental Figure 1 (see panels S1A-S1F and S1H-S1J).

2. “The data in Figures 4 and 5 provide compelling data regarding the impact of aberrant CXCR4 signaling on osteogenic specification at the level of transcriptional effects and cell cycle progression. The PCA data shown in Figure 4C is not well explained as the 48 genes used for expression profiling are not described in the text nor the supplement, which lists a smaller number of genes. The data in the subsequent panels are more informative. I would consider removing panel 4C or moving it to the supplement with a better description of the analysis. The experiments shown in Figure 5 document the selective reduction in osteogenic differentiation capacity of stromal stem cells carrying one or two mutant CXCR4 receptors in a dose dependent fashion and the reversal of this phenotype with treatment of the receptor inhibitor AMD3100. The relevance of these data in mice to human bone biology are supported with the data shown in Figure 6 which revealed a selective osteogenic differentiation defect in bone marrow cells derived from WHIM syndrome patients.”

We thank the reviewer for bringing to our attention that the PCA data shown in Figure 4C was not clear and we apologize for that. This panel has now been removed. As suggested by Reviewer#1, we decided to investigate the impact of the gain-of-Cxcr4-function on the molecular identity of SSCs by RNA-seq analyses of sorted bulk cells from WT and mutant bone fractions. Biological processes related to cell cycle and osteogenic differentiation were significantly modulated in

1013/1013 SSCs as determined by GSEA (Gene set enrichment analysis) (see new Figure 4C). The SSC signature in 1013/1013 mice was reduced for genes related to cell cycle progression and regulation (see new Figures S2A and S2B). Likewise, genes related to osteogenic differentiation appeared to be decreased in mutant SSCs (see new Figures 4D and 4E). In contrast, key genes involved in both adipogenesis and chondrogenesis were not differentially expressed in mutant SSCs (see new Figure S2C). These results were confirmed by microfluidic-based multiplex gene expression analyses (see Figures 4F and 4G and Figure S2D), thus suggesting that proper Cxcr4 signaling is required for regulating osteogenic specification of SSCs at the transcriptional level. The text has been modified accordingly (page 13, line 264).

3. “With regards to the conclusion that a skeletal phenotype is present in a subset of WHIM syndrome patients, given that treatment of neutropenia with G-CSF is associated with osteopenia as side effect of therapy, it would be useful to know the total number of patients treated with G-CSF in the cohort to address the concern that the enrichment in osteopenic patients is restricted to those patients that have been so treated as well as their ages and genders given the impact of these variables on risk for osteopenia in general.”

We thank the reviewer for this very relevant comment. Nineteen WS patients had a baseline bone density scan as part of a drug treatment trial (NCT02231879) comparing 1 year of twice daily filgrastim (Neupogen) versus plerixafor (Mozobil) in a randomized, blinded crossover design. There were 13 women and 6 men with an average age of 30.5 years (range 10-56). Patients had been on filgrastim for an average of 5.7 years prior to enrolling in the trial (range 0-27). Six of the 19 had not used filgrastim regularly prior to trial enrollment. These findings suggest that the enrichment in osteopenic WS patients is not merely due to treatment regimen, age or gender parameters. This point is now mentioned in the revised version of the manuscript (page 22, line 487; page 23, line 507).

4. “Apart from these concerns, the quality of the data presented is good and the conclusions supported by the evidence. The manuscript is well written and the references appropriate, though it was notable that the initial description of the cause of WHIM syndrome as gain-of-function truncation mutations in CXCR4 was not cited, this should be added.”

We thank the reviewer for this relevant comment and apologize for this oversight. The initial description of inherited *CXCR4* mutations in the WS has been reported by Hernandez and collaborators in 2003 (Nature Genetics, PMID: 12692554). The appropriate reference (n°50 in the list of references) has been added accordingly (page 5, line 98).

REVIEWER COMMENTS

Reviewer #1 (Remarks to the Author):

The authors answered adequately to the comments and questions by performing additional experiments and adapting the text. The claims are now well supported by their findings, making it an interesting study providing further insight in the skeletal effects of CXCR4 mutations.

Minor comments:

Page 11, line 208: the following sentence is difficult to interpret: Cxcr41013-bearing mice exhibited unremarkable bone formation. Not clear what is meant by 'unremarkable'.

Page 11, line 225: It is mentioned that OBL differentiation is reduced in mutant mice, whereas the previous lines describe normal osteogenic differentiation when cultures are started with OPCs. To avoid misunderstanding, some other wording should be used to describe that the transition of SSCs to OPCs is impaired or that there is reduced osteogenic lineage commitment.

Reviewer #2 (Remarks to the Author):

The authors have given a satisfactory response to some of this reviewer's concerns, improving the manuscript. However, their answers to several issues remain incomplete, and therefore their conclusions are still not convincing.

1. The new data that transplantation of Cxcr4 1013/1013 mutant bone marrow cells markedly reduced trabecular bone content (BV/TV and Tb.Nb) of wild-type recipient mice (Fig. 2P) are interesting and important. The magnitude of the decrease seems to be much larger compared with Cxcr4 1013/1013 mutant mice, suggesting that microenvironments with gain-of-function Cxcr4 1013/1013 mutations increased and rescued trabecular bone content. This is consistent with previous findings that deletion of CXCR4 in mesenchymal cells reduced trabecular bone content (Tzeng et al., J. Bone Miner. Res. 2018; Zhu et al., J. Biol. Chem. 2011).

2. Again I would recommend the authors to generate and analyze the mice, in which mesenchymal cells, including SSCs, but not hematopoietic cells carry WS-linked CXCR4 mutations. However, the authors mentioned that it was not feasible in the frame of a reviewing period. Then, the authors should at least show trabecular bone content (BV/TV and Tb.Nb) of Cxcr4 1013/1013 mutant mice transplanted with bone marrow cells from wild-type mice and compare the results with those of wild-type mice transplanted with mutant bone marrow cells.

Reviewer #4 (Remarks to the Author):

The authors have addressed all of the issues raised by me adequately. I am satisfied with the responses to the other reviewers as well and have no additional concerns.

Point-to-point response to the reviewers' comments

Reviewer #1 comments:

The authors answered adequately to the comments and questions by performing additional experiments and adapting the text. The claims are now well supported by their findings, making it an interesting study providing further insight in the skeletal effects of CXCR4 mutations.

Minor comments:

1. Page 11, line 208: the following sentence is difficult to interpret: Cxcr41013-bearing mice exhibited unremarkable bone formation. Not clear what is meant by 'unremarkable'.

We thank the reviewer for bringing to our attention that the use of the term "unremarkable" was not clear and probably not appropriate. The sentence has been modified as follow: "Cxcr4¹⁰¹³-bearing mice exhibited similar bone formation as revealed by osteoid surface (OS/BS) and osteoblast surface (Obl.S/BS) compared to WT mice (Fig. 3C)". The text has been modified accordingly (page 11, line 208).

2. Page 11, line 225: It is mentioned that OBL differentiation is reduced in mutant mice, whereas the previous lines describe normal osteogenic differentiation when cultures are started with OPCs. To avoid misunderstanding, some other wording should be used to describe that the transition of SSCs to OPCs is impaired or that there is reduced osteogenic lineage commitment.

We thank the reviewer for this very relevant comment. The sentence has been changed as follow: "These findings suggest reduced osteogenic lineage commitment in Cxcr4¹⁰¹³-bearing mice". The text has been modified accordingly (page 11, line 224).

Reviewer #2 comments:

The authors have given a satisfactory response to some of this reviewer's concerns, improving the manuscript. However, their answers to several issues remain incomplete, and therefore their conclusions are still not convincing.

1. The new data that transplantation of Cxcr4 1013/1013 mutant bone marrow cells markedly reduced trabecular bone content (BV/TV and Tb.Nb) of wild-type recipient mice (Fig. 2P) are interesting and important. The magnitude of the decrease seems to be much larger compared with Cxcr4 1013/1013 mutant mice, suggesting that microenvironments with gain-of-function Cxcr4 1013/1013 mutations increased and rescued trabecular bone content. This is consistent with previous findings that deletion of CXCR4 in mesenchymal cells reduced trabecular bone content (Tzeng et al., J. Bone Miner. Res. 2018; Zhu et al., J. Biol. Chem. 2011).

We thank the reviewer for these valuable remarks and suggestions. Although it is difficult to compare chimeric and steady state mice, especially considering the whole-body irradiation and the 4-month reconstitution period, it appears indeed that WT mice reconstituted with mutant BM display stronger trabecular bone defects than younger mutant mice at steady state. It is indeed possible that, as put by the reviewer, "microenvironments with gain-of-function Cxcr4 1013/1013 mutations increased and rescued trabecular bone content". Besides mesenchymal and osteolineage cells, the BM ecosystem contains other possible effector cells including hematopoietic and mature immune cells, but also some radioresistant endothelial cells, and other stromal cells such as adipocytes that all express the CXCR4 receptor, which emerge during bone development and reach homeostasis at the adult stage. One can assume that the gain of CXCR4 function might modulate, positively or negatively, one, or several, BM landscape component(s) that could balance the trabecular bone defect in 1013/1013 mice. This is indeed consistent with

previous works reporting a cell-autonomous Cxcl12-Cxcr4 signaling on the MSPC osteogenesis but not adipogenesis (Tzeng et al., J. Bone Miner. Res. 2018; Zhu et al., J. Biol. Chem. 2011). Future studies are necessary to identify such cells by tissue-specific CXCR4 targeting, as suggested by the reviewer. This point has been now discussed (page 21, line 469).

The magnitude of the bone loss in WT recipient upon transplantation of mutant BM is surprising but is indicative of cell-extrinsic hematopoietic-driven Cxcr4-mediated regulation of the skeletal landscape. This may be due to the presence of effector mature leukocytes in the BM graft as we suggested previously in response to the point 2 raised by Reviewer#1. Indeed, one can speculate that myeloid cells including OCL progenitors as well as lymphoid cells may actively participate in promoting bone remodeling in BM chimeric WT recipient mice. Indeed, the laboratory of Pr. A. Bozec among others recently reported that prolonged HIF-1 α signaling in B cells leads to enhanced RANKL production and OCL formation in the BM (Meng et al., Bone Research 2022). Likewise, BM T cells are known to produce RANKL and to regulate OCL compartment within the BM (see for review for instance Corrado et al., IJMS 2020; Mori et al., Clin Dev Immunol 2013; Zhang et al., Front Endocrinol 2020). Whether the transplantation of mutant BM recreates a pro-osteoclastogenic environment through a remodeling of the lymphoid compartment deserves further investigations.

2. Again I would recommend the authors to generate and analyze the mice, in which mesenchymal cells, including SSCs, but not hematopoietic cells carry WS-linked CXCR4 mutations. However, the authors mentioned that it was not feasible in the frame of a reviewing period. Then, the authors should at least show trabecular bone content (BV/TV and Tb.Nb) of Cxcr4 1013/1013 mutant mice transplanted with bone marrow cells from wild-type mice and compare the results with those of wild-type mice transplanted with mutant bone marrow cells.

We are grateful to the reviewer for this helpful suggestion and as requested, we have quantified trabecular bone parameters by μ CT (see new Figures 2H and 2I). These new analyses extend the flow-cytometric and histological ones and indicate a persistent bone loss in mutant recipient upon transplantation of WT BM, thereby supporting the hypothesis of a cell-intrinsic CXCR4-mediated regulation of the skeletal landscape. However, the extent of the bone loss appears to be less marked compared with WT recipients transplanted with mutant BM cells. As discussed above, this could rely on the modulatory effect due to the gain of CXCR4 function on one, or several, stromal component(s) that could compensate the trabecular bone defect particularly in 1013/1013 mice. As stated in the manuscript, these findings suggest that impaired CXCR4 desensitization in both skeletal and hematopoietic cells have combinatorial effects on bone landscape dysregulation in adult *Cxcr4*¹⁰¹³-bearing mice.

Reviewer #4 comments:

The authors have addressed all of the issues raised by me adequately. I am satisfied with the responses to the other reviewers as well and have no additional concerns.

We thank the reviewer for the previous suggestions and are happy that all concerns were addressed.

REVIEWERS' COMMENTS

Reviewer #2 (Remarks to the Author):

The authors have given a satisfactory response to this reviewer's concerns, improving the manuscript.

Minor points

Figure 2I: How about p-values in Tb.Nb? They appear to be significantly less than 0.005 as seen in BV/TV.

Point-to-point response to the reviewers' comments

Reviewer #2 comments:

The authors have given a satisfactory response to this reviewer's concerns, improving the manuscript.

Minor points

Figure 2I: How about p-values in Tb.Nb? They appear to be significantly less than 0.005 as seen in BV/TV.

We thank the reviewer for his/her previous constructive comments that have helped us improving greatly the quality of our manuscript. We are glad to read that all concerns were now addressed. Regarding the p-values of the data shown in Figure 2I, they have been determined using the two-tailed Student's *t* test and are as follow:

- For the BV/TV parameter:

WT BM-chimeric CD45.2+ WT vs WT BM-chimeric CD45.2+ +/-1013 mice: P= 0.033

WT BM-chimeric CD45.2+ WT vs WT BM-chimeric CD45.2+ 1013/1013 mice: P = 0.0244

- For the Tb.Nb parameter:

WT BM-chimeric CD45.2+ WT vs WT BM-chimeric CD45.2+ +/-1013 mice: P= 0.0702

WT BM-chimeric CD45.2+ WT vs WT BM-chimeric CD45.2+ 1013/1013 mice: P = 0.0710

The exact p-values have been now provided in the legend of Figure 2.